# Neutral evolution of snoRNA Host Gene long non-coding RNA affects cell fate control

Matteo Vietri Rudan [1], Kalle H Sipilä[1], Christina Philippeos [1], Clarisse Ganier[1], Priyanka G Bhosale [1], Victor A Negri[1] & Fiona M Watt [1,2]✉

## Abstract

**A fundamental challenge in molecular biology is to understand how evolving genomes can acquire new functions. Actively transcribed, non-coding parts of the genome provide a potential platform for the development of new functional sequences, but their biological and evolutionary roles remain largely unexplored. Here, we show that a set of neutrally evolving long non-coding RNAs (lncRNAs) whose introns encode small nucleolar RNAs (snoRNA Host Genes, SNHGs) are highly expressed in skin and dysregulated in inflammatory conditions. Using SNHG7 and human epidermal keratinocytes as a model, we describe a mechanism by which these lncRNAs can increase self-renewal and inhibit differentiation. The activity of SNHG7 lncRNA has been recently acquired in the primate lineage and depends on a short sequence required for microRNA binding. Taken together, our results highlight the importance of understanding the role of fast-evolving transcripts in normal and diseased epithelia, and show how poorly conserved, actively transcribed non-coding sequences can participate in the evolution of genomic functionality.**

**Keywords** lncRNA; Evolution; Skin
**Subject Categories** Evolution & Ecology; RNA Biology; Skin

## Introduction

Advances in the annotation of the genome have yielded the unexpected finding that non-coding sequences are pervasively transcribed (Djebali et al, 2012; Abascal et al, 2020). While this could imply widespread function (Jandura and Krause, 2017; Boland, 2017), some studies have highlighted that the emergence of non-functional and/or redundant sequences is, rather, a by-product of genomic evolution (Palazzo and Koonin, 2020; Ohno, 1999).

To distinguish the functional from the non-functional part of the genome the most robust and most commonly used parameter is evolutionary conservation, which in fact is used as the defining characteristic of biological function in its narrower sense (Doolittle et al, 2014). However, use of sequence conservation alone is not sufficient to capture the full extent of genome functionality (Ponting, 2017); recent studies have highlighted how some poorly conserved genomic regions, either regulatory or transcribed, can play important roles in the development of new biological features, or contribute to the phenotypic diversity among species, particularly in the primate lineage (Luo et al, 2021; Mangan et al, 2022; Bi et al, 2023). Long non-coding RNAs (lncRNAs) are a class of transcripts that generally exhibit a lower degree of sequence conservation when compared to protein-coding or small non-coding RNAs. Such conservation can be limited to only a portion of the sequence, to the intron/exon structure or simply to the gene's position within the genome (Ulitsky, 2016). In spite of this, several lncRNAs have been ascribed a wide variety of functions through many different mechanisms (Statello et al, 2021). While we have accumulated considerable knowledge about the biological roles of a few non-coding RNAs, in most cases their function, if any, remains elusive (Ponting and Haerty, 2022).

Among the most evolutionarily dynamic tissues is the epidermis, a multi-layered epithelium predominantly composed of keratinocytes that forms the outermost surface of the body. The epidermal basal layer contains stem cells that maintain the tissue and can either undergo cell division or begin a process of terminal differentiation, during which they migrate towards the surface of the skin, through the spinous and granular layers, and form the protective covering surface of the organism, the cornified layer (Fuchs and Raghavan, 2002). Control of the equilibrium between self-renewal and differentiation of keratinocytes is essential for the correct maintenance and repair of the epidermis (Cangkrama et al, 2013; Sipilä et al, 2022). Due to its position at the interface with the external environment, the epidermis is a rapidly evolving, adaptable tissue, as evidenced, for example, by the relatively high rate of amino-acid substitutions in epidermal proteins, underlying a diverse array of phenotypes between closely related species or even between different human populations (Brettmann and Strong, 2018). While a few RNA regulators of cell fate have been described in the epidermis (Kretz et al, 2012, 2013; Zhang et al, 2023), the contribution of non-coding parts of the genome to such evolutionary plasticity has received little attention.

Here, we have focussed on small nucleolar RNA Host Genes (SNHGs), a set of highly expressed, extremely poorly conserved lncRNAs whose levels are dysregulated in skin diseases. By using

[1]Centre for Gene Therapy and Regenerative Medicine, King's College London, Floor 28, Tower Wing, Guy's Hospital, Great Maze Pond, London SE1 9RT, UK. [2]Present address: Directors' Unit, EMBL, Meyerhofstr. 1, 69117 Heidelberg, Germany. ✉E-mail: fiona.watt@embo.org

primary human epidermal keratinocytes as a well-established model, we demonstrate the function of SNHGs in regulating the balance between self-renewal and differentiation. Furthermore, comparison of SNHG activity in keratinocytes from multiple mammalian species across different evolutionary distances provides insights into how new functions can be acquired by the genome.

# Results

## SNHGs are a class of poorly conserved, highly abundant long non-coding RNAs that are expressed in healthy and inflamed epidermis and can affect keratinocyte fate

The functional contribution of the major evolutionarily conserved signalling pathways in normal and diseased epidermis has undergone extensive investigation. In contrast, the biological role of less conserved transcripts has not been widely investigated, even though they can be strongly expressed in normal tissue and altered in skin diseases. To identify potentially functional poorly conserved transcripts in the epidermis, we analysed keratinocyte single-cell RNA sequencing (scRNAseq) data (Reynolds et al, 2021). We compared cells from unaffected skin with matched atopic dermatitis (AD) and psoriasis lesions and estimated the conservation of significantly differentially expressed genes using average PhastCons scores from multiple alignments of 100 vertebrate genomes. While many of the affected transcripts were highly conserved coding genes, we were able to identify several very poorly conserved transcripts that were altered in one or both pathological states. As the degree of conservation decreased, we could see an increasing number of lncRNA species, among which we noticed several relatively highly expressed SNHGs (Fig. 1A,B), a class of lncRNA that contain small nucleolar RNAs (snoRNAs) within their introns.

Generally, lncRNAs are expressed at low levels. However, in the case of SNHGs, the production of the abundant intronic snoRNAs requires transcription of the host gene. Indeed, when compared to other transcript classes genome-wide, SNHGs peculiarly display a very low degree of conservation, even relative to known functional lncRNAs (Fig. S1A) yet retain a level of expression in the same range as most protein-coding genes, as evaluated by using the GTEx scores from data generated by the Gene-Tissue Expression project (Fig. 1C,D; Appendix Fig. S1B). This broadly high expression level of SNHGs appears to be conserved, as a similar trend is also observed when comparing the expression of different gene classes in mouse genome-wide data from Tabula Muris (Appendix Fig. S1C).

SNHG lncRNA sequences contain a relatively high GC content when compared to other classes of genes (Appendix Fig. S1B). SNHG promoters also appear to be less conserved than coding gene promoters (Fig. S1C). To control for the possible influence of the phylogenetic scale on our results, we repeated our conservation analyses using average PhastCons scores from multiple alignments of 30 mammals (28 primates) but saw no substantial differences in the outcomes (Appendix Fig. S1D–F). We also assessed the rate at which SNHG lncRNAs are evolving. We used PhyloP scores (Pollard et al, 2010) to gauge what percentages of the exonic sequence of each SNHG are under purifying selection (PhyloP > 2), are evolving neutrally ($-2 \leq$ PhyloP $\leq 2$) or are experiencing an accelerated rate of change (PhyloP < $-2$). As expected, conserved nucleotides make up a very small fraction of each transcript and while in some cases evidence of accelerated evolution can be observed in up to 8% of base pairs, the vast majority of the sequence of SNHG lncRNAs is evolving neutrally (Fig. 1E). It should be noted that while SNHGs are broadly expressed, there is variation in their levels across tissues, which suggests some degree of regulation of their abundance (Appendix Fig. S2).

As atopic dermatitis and psoriasis lesions are characterised by an imbalance between self-renewal and differentiation of epidermal cells, we sought to understand if SNHGs could be involved in the regulation of keratinocyte differentiation in normal tissue. We clustered scRNAseq data of keratinocytes from healthy skin donors (Reynolds et al, 2021) into different states corresponding to stages of differentiation, as described previously (Negri and Watt, 2022; Negri et al, 2023) (Fig. 2A). We found 13 SNHGs expressed at various levels at multiple differentiation stages (Fig. 2B). We selected the five most highly expressed keratinocyte SNHGs whose expression was also significantly changed in AD or psoriasis (SNHG7, 8, 12, 15, and 19, Fig. S3A) for further investigation. Expression of these SNHGs could be detected across all clusters, although some enrichment was seen in keratinocytes transitioning from basal to more differentiated cell states, with different SNHGs having maximal expression in either of the two transition states or in certain spinous clusters (Fig. 2C). We also used scRNAseq data to estimate the number of SNHG molecules per cell (Appendix Fig. S3B). We found SNHG expression levels to be in the same range as key epidermal transcription factors such as p63. It should be noted that these estimated counts will be dependent on the depth of sequencing and can thus represent an underestimate of the actual number of molecules in each cell.

A variety of signalling pathways participate in regulating the balance between self-renewal and differentiation of keratinocytes (Janes et al, 2004, 2009; Connelly et al, 2011; Mishra et al, 2017; Hiratsuka et al, 2020). The variation in expression of SNHGs during differentiation suggests that they might be under the control of different pathways. Indeed, treatment with a panel of pathway inhibitors had differential effects on the expression of each SNHG as quantified by qPCR. Inhibition of the phosphoinositide-3-kinase (PI3K)/RACα serine/threonine-protein kinase (Akt) pathway increased the levels of SNHG7 and SNHG8, and the latter was also significantly increased after Protein Kinase C or translation inhibition. SNHG12 levels increased upon Focal Adhesion Kinase inhibition, but decreased when the mitogen-activated kinase MEK was blocked. SNHG15 levels decreased in response to Akt or MEK inhibition, while levels of SNHG19 were reduced when the cells were treated with inhibitors of Akt or translation (Fig. 2D; Appendix Fig. S4).

In light of these results, we tested whether the presence of SNHG transcripts might have any functional relevance. We used siRNA to knock down each of five highly expressed SNHGs in primary human keratinocytes and performed a clonogenicity assay as a readout of their self-renewal potential (Mishra et al, 2017; Barrandon and Green, 1987). Remarkably, a reduction in the expression level of any of the SNHGs caused a significant decrease in clonogenicity (Fig. 2E; Appendix Fig. S5), indicating that neutrally evolving non-coding transcripts arising from snoRNA host genes can affect the balance between self-renewal and proliferation.

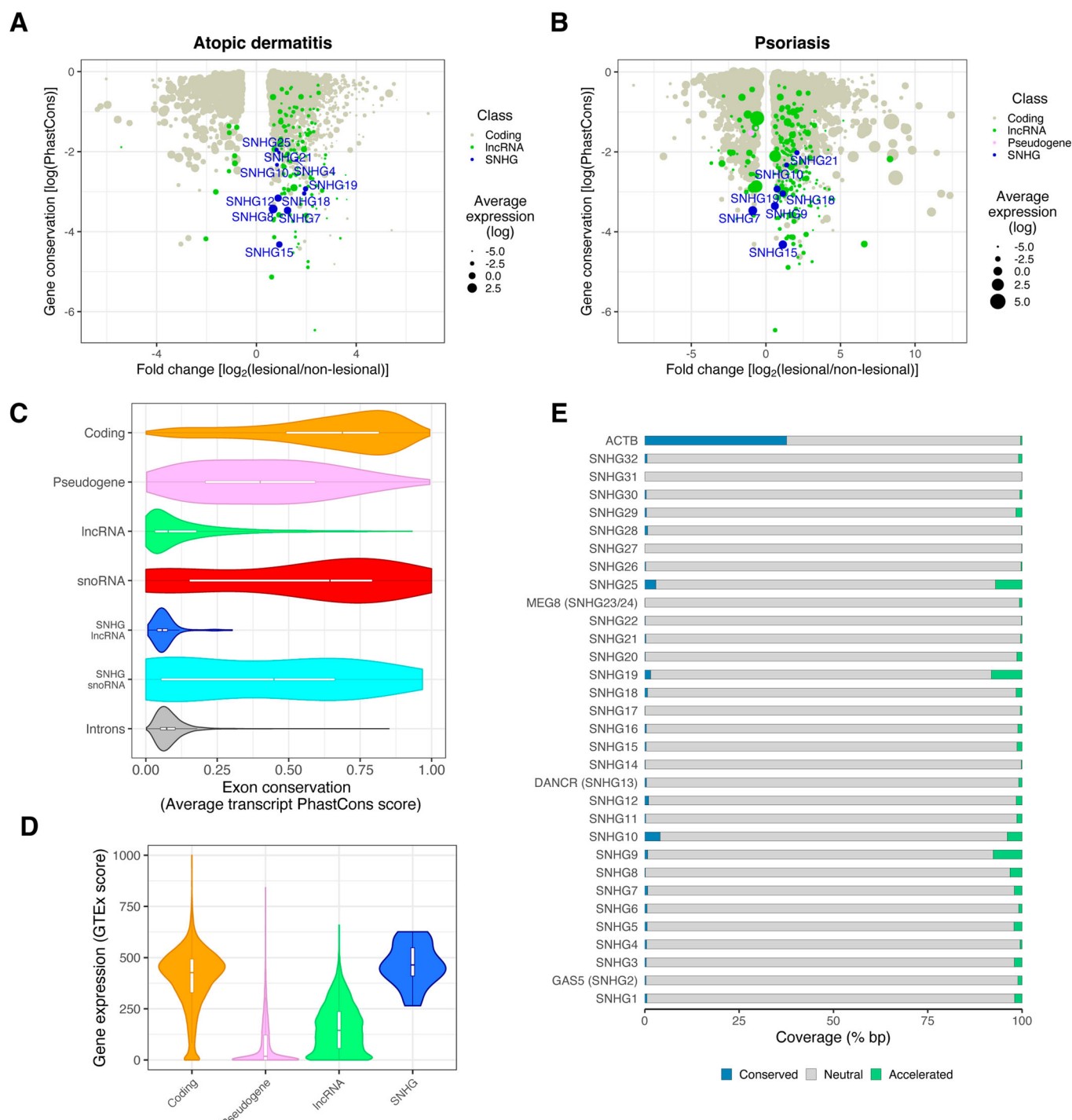

**Figure 1. SNHGs encode highly expressed long non-coding RNAs undergoing neutral evolution that are regulated in skin inflammatory conditions.**

Landscape of significantly ($p_{adj}$ < 0.01) differentially expressed genes relative to their conservation in (**A**) atopic dermatitis and (**B**) psoriasis. Distributions of (**C**) exonic sequence conservation and (**D**) gene expression scores of SNHG lncRNA compared to other classes of transcripts genome-wide. Boxplots within the violins indicate the median and the interquartile range. (**E**) Rates of evolution of the nucleotides in SNHG exonic sequences assessed by PhyloP scores. Beta-actin (ACTB) is shown as a reference for the rates of evolution in a typical protein-coding gene. Source data are available online for this figure.

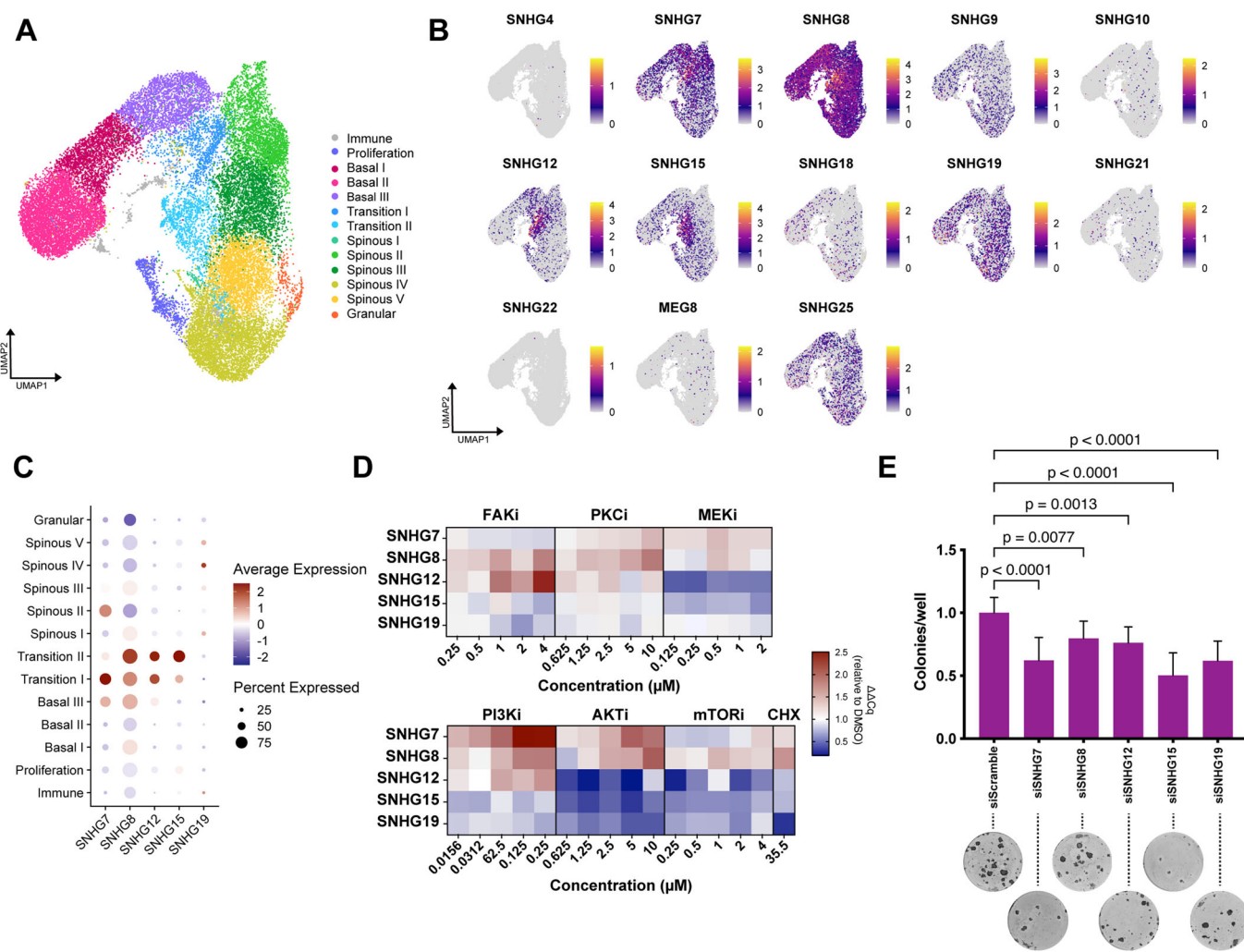

**Figure 2. SNHG expression is regulated in epidermal keratinocytes and can impact their fate.**

(A) UMAP clustering of single-cell RNA sequences of epidermal keratinocytes. Clusters are shown in different colours and labelled based on marker gene expression. Reproduced from Negri et al (2023). (B) Single-cell level expression of the SNHGs detectable in the dataset overlayed on the UMAP clusters. (C) Average expression and percentage of keratinocytes expressing five selected SNHGs among the clusters. (D) SNHG expression in response to inhibition of multiple signalling pathways. For each inhibitor the change in expression of five epidermally expressed SNHGs compared to DMSO control is shown after 4 h incubation with increasing concentrations of the inhibitor. (E) Clonogenicity assay of keratinocytes treated with either non-targeting control siRNA (siScramble) or siRNAs targeting selected epidermally expressed SNHG ncRNAs. Data shown are mean +/− SD. Ordinary one-way ANOVA and Dunnett's multiple comparisons test, $n = 12$ wells. Source data are available online for this figure.

## SNHG7 lncRNA promotes self-renewal and prevents differentiation of keratinocytes

Next, we focused on SNHG7 since it is highly expressed in the epidermis (Fig. 2B,C; Appendix Fig. S2), its expression is significantly changed in both AD and psoriasis (Fig. 1A,B), it strongly regulates clonogenicity (Fig. 2E), and its gene structure is relatively simple: SNHG7 encodes only two alternative lncRNAs (Dunham et al, 2012; Boone et al, 2020), each about 1 kb long, and contains two H/ACA box snoRNA genes, SNORA17A and SNORA17B, within its introns (Fig. 3A).

The modulation of SNHG7 lncRNA transcription is linked to expression of the snoRNAs hosted in its introns. Our screen of pathway inhibitors showed that SNHG7 was markedly upregulated upon PI3K inhibition (Fig. 2E; Appendix Fig. S4). Intriguingly, this

was not mirrored by the levels of a snoRNA hosted by SNHG7 (Fig. 3B). When we inhibited transcription with Actinomycin D before blocking PI3K, a significant increase in SNHG7 could still be detected, while the transcription-dependent, PI3K-mediated up-regulation of ERBB3 (Chakrabarty et al, 2012) was completely prevented (Fig. 3C; Appendix Fig. S6A). These results point to the presence of a post-transcriptional regulatory mechanism for the lncRNA, allowing its expression to be decoupled from the snoRNA.

In vivo, SNHG7 was present in scattered cells throughout the living layers of human epidermis, with higher expression in the basal and spinous layers (Fig. 3D) as shown by single-molecule RNA fluorescence in situ hybridisation (smRNA FISH), and in agreement with our scRNAseq analysis. In non-homeostatic conditions, we observed alterations in SNHG7 lncRNA levels. SmRNA FISH of AD lesions showed an increase in SNHG7

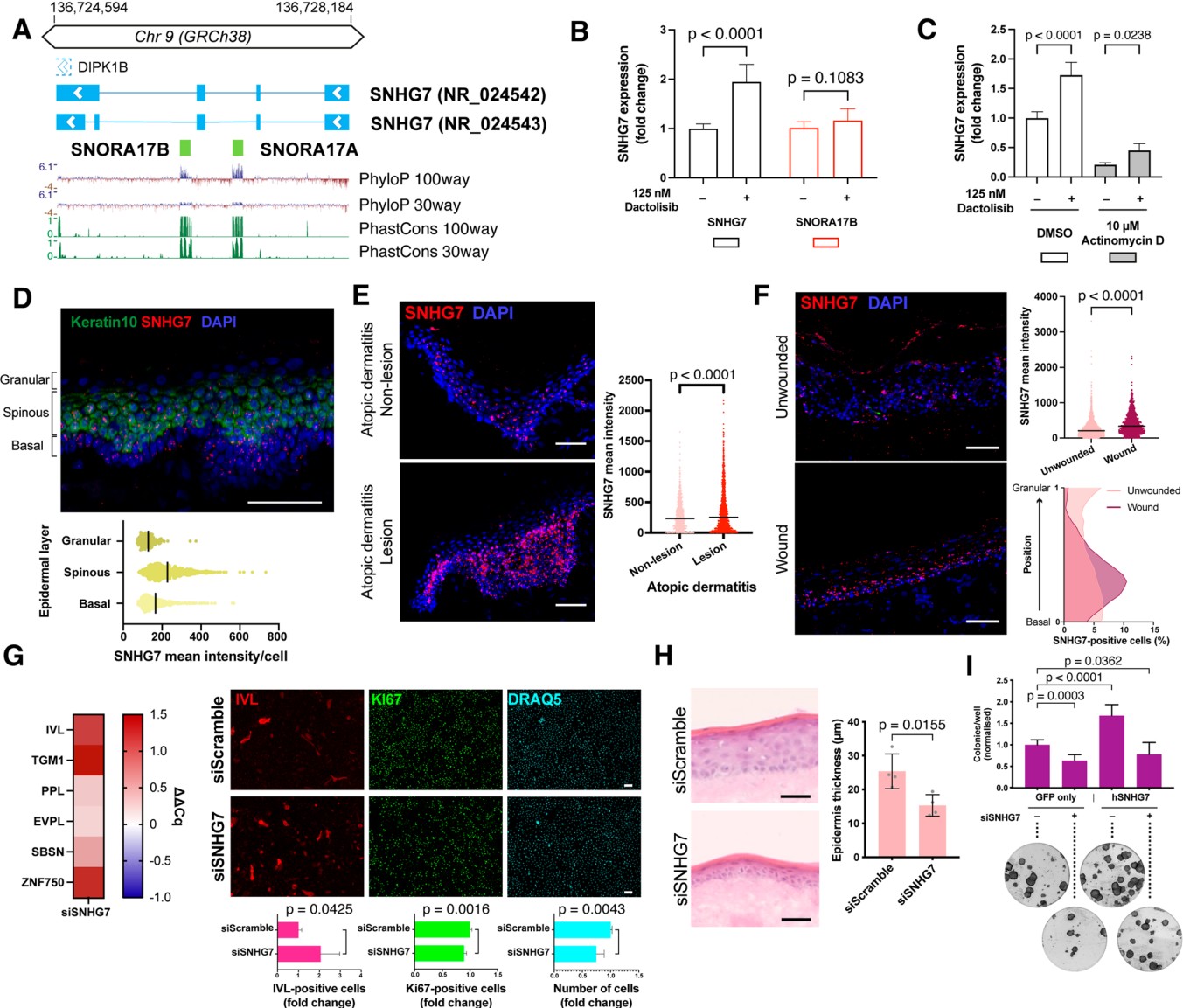

**Figure 3. Epidermal SNHG7 lncRNA expression is modulated in homeostatic and non-homeostatic conditions and is necessary to maintain proliferation and prevent differentiation.**

(A) Schematic representation of the SNHG7 genomic locus. Thick lines represent exons, while the thinner lines joining them represent introns. Arrows indicate the direction of transcription. An overlapping 5′UTR of the neighbouring DIPK1B gene is shown as a dashed outline. Conservation tracks are displayed below the schematic. (B) Expression of SNHG7 and SNORA17B after 4 h treatment with PI3K inhibitor Dactolisib. $n = 9$ independent treatments. (C) SNHG7 expression after 4 h PI3K inhibition with or without 1 h prior treatment with transcription inhibitor Actinomycin D. $n = 5$ independent treatments. (D) smRNA FISH of SNHG7 lncRNA in healthy human facial skin. Keratin 10 RNA staining was used to mark the spinous layer and the expression level of SNHG7 was quantified in each epidermal layer. (E) smRNA FISH of SNHG7 lncRNA in sections of lesional and non-lesional atopic dermatitis skin, with expression quantification. $n > 600$ cells. (F) smRNA FISH of SNHG7 lncRNA in re-epithelialised wounded biopsies. Shown are sections of the areas outside the wound ('Unwounded') as well as the newly epithelialised region ('Wound'). Plots on the right show quantification of expression levels in the different areas (top) and the distribution of SNHG7-espressing cells in the different epidermal layers (bottom). (G) Effect of SNHG7 knockdown on keratinocyte differentiation and proliferation. Left panels, heatmap of expression changes of multiple differentiation markers in knocked down cells 72 h post-transfection. Right panels, staining of differentiation (IVL) and proliferation (Ki67) markers 96 h post-transfection, with quantification. $n \geq 4$ independent transfections. (H) Effect of SNHG7 knockdown on the ability of keratinocytes to form a new epithelium. $n = 4$ DEDs. (I) Rescue of the SNHG7 knockdown phenotype by lncRNA overexpression. Clonogenicity assay quantification with representative images. Scale bars, 50 μm. Data shown in all bar plots are mean $+/-$ SD. Lines in dot plots indicate the median. (B, G, H) Two-tailed unpaired t-test. (C) Ordinary one-way ANOVA and Šidàk's multiple comparisons test. (E, F) Kolmogorov–Smirnov test. (I) Ordinary one-way ANOVA and Dunnett's multiple comparison test. Source data are available online for this figure.

expression (Fig. 3E), which was consistent with scRNAseq from AD patients.

To experimentally demonstrate the modulation of SNHG7 in non-homeostatic conditions, we used ex vivo wounding of human skin explants. The re-epithelialising keratinocytes exhibited increased levels of SNHG7 lncRNA, including more prominent expression in basal cells (Fig. 3F). This supports the idea that SNHG7 participates in regulating the balance between self-renewal and differentiation in the context of epidermal repair. Our results also indicate that keratinocyte SNHG7 expression can increase in response to perturbation of homeostasis in the absence of circulating immune cells.

To further characterise the reduction in clonogenicity observed upon downregulation of SNHG7, we stained keratinocytes with antibodies to the proliferation marker Ki-67 and the differentiation marker Involucrin (IVL). We observed a reduced rate of proliferation and an increased percentage of differentiated cells upon SNHG7 knockdown. The induction of differentiation was confirmed by the increase in mRNA levels of a range of differentiation markers (Fig. 3G). Knockdown cells also displayed a reduced ability to form an epidermis on de-epidermised dermal substrates (DEDs, Fig. 3H). We did not detect significant alterations in the distribution of epidermal markers in the reconstituted tissues (Appendix Fig. S6B). Off-target effects were excluded by deconvolution of the siRNA pools used in our experiments (Appendix Fig. S6C,D). The effects of SNHG7 knockdown were not due to induction of apoptosis, since there was no increase in the number of cells expressing cleaved caspase 3 (Appendix Fig. S6E).

We also ruled out any potential contribution of the snoRNA genes hosted within SNHG7 to the knockdown phenotype. Firstly, no reduction in protein translation as assessed by the OP-Puro assay was seen in knockdown cells (Appendix Fig. S6F). Secondly, consistent with the post-transcriptional nature of RNA interference, siRNA-mediated knockdown of SNHG7 only affected the exonic portion of the gene (Appendix Fig. S6G). We further demonstrated that the effects we observed were due to the deficiency in SNHG7 lncRNA by overexpressing its spliced form (NR_024542). Since the siRNA is able to target both the endogenous and the exogenous lncRNA we first identified the minimum siRNA concentration necessary to see a clear phenotype (Appendix Fig. S6H). Overexpression of SNHG7 lncRNA led to a nearly complete rescue of the knockdown phenotype (Fig. 3I; Appendix Fig. S6I–K).

Overexpression of SNHG7 caused a significant increase in clonogenic potential (Fig. 3I). In addition, stable overexpression of SNHG7 allowed the keratinocytes to be kept in culture for seven additional passages before losing their self-renewal capacity (Fig. S6L). This, together with the high levels of expression of SNHG7 during keratinocyte commitment and early differentiation (Fig. 2C), prompted us to investigate whether SNHG7 might affect suspension-induced differentiation of disaggregated human keratinocytes (Mishra et al, 2017). While we did observe a more rapid induction of the pro-commitment phosphatase DUSP6 upon SNHG7 knockdown (Appendix Fig. S6M), the kinetics of differentiation in suspension were largely unchanged. In addition, overexpression of SNHG7 did not prevent the loss of colony forming ability with time in suspension (Appendix Fig. S6N). These results indicate that SNHG7 influences the propensity of

keratinocytes to undergo self-renewal, but its activity can be overridden by the strong differentiation stimulus of detachment from the extracellular matrix.

## SNHG7 function has evolved recently in the primate lineage

The low sequence conservation of SNHGs raises the possibility that their effects on the regulation of cellular fate can be acquired over relatively short evolutionary timescales. However, cases have been described of lncRNAs which maintained their function across large evolutionary distances, in the absence of widespread sequence conservation (Ulitsky et al, 2011). In order to distinguish between these alternative scenarios, we assessed the biological activity of SNHG7 in night monkey (*Aotus trivirgatus*) and mouse (*Mus musculus*) primary keratinocytes. Given the absence of genomic sequence information for *A. trivirgatus*, we first performed 5′ and 3′ Rapid Amplification of cDNA ends (RACE) followed by sequencing of the SNHG7 lncRNA isolated from the night monkey cells. We could identify at least two transcripts, both of which displayed relatively high sequence similarity to the human lncRNA in their 3′ regions (Fig. 4A). Conversely, the murine Snhg7 sequence retains little homology to the human lncRNA throughout its sequence (Fig. 4A). Inhibition of PI3K increased the levels of SNHG7 in night monkey cells, as in human, while it did not affect its expression in mouse keratinocytes (Fig. 4B).

Although SNHG7 expression in night monkey cells was regulated by PI3K, the colony formation ability of night monkey and mouse keratinocytes was not affected by SNHG7 knockdown (Fig. 4B,C; Appendix Fig. S7A,B). This indicates a rapid acquisition of functionality to accompany the variation in sequence. We therefore sought to investigate whether human SNHG7 was sufficient to affect the proliferation/differentiation balance in non-human keratinocytes. To this end, we stably expressed the human SNHG7 lncRNA in night monkey and mouse keratinocytes. Human SNHG7 lncRNA was able to increase the colony forming capacity of night monkey keratinocytes but had no effect on the mouse cells (Fig. 4B,C; Appendix Fig. S7B,C).

Our data therefore point to the presence of elements in the human SNHG7 sequence that are able to influence cell fate, but whose efficacy appears conditional on the presence of additional factors and/or conditions within their cellular environment.

## Knockdown of SNHG7 leads to downregulation the targets of interacting miRNAs

Next, we sought to characterise the molecular mechanism of SNHG7's action. RNAseq of primary human keratinocytes at 24 and 48 h after SNHG7 knockdown (Appendix Fig. S8A) revealed that the differentially regulated genes were predominantly involved in control of proliferation, cell motility and cell adhesion (Fig. 5A,B). Already 24 h after transfection, the differentially expressed genes exhibited a notable bias towards downregulated genes (Fig. 5A), which is consistent with the idea that SNHG7 might be a miRNA regulator, as reported in cancer settings (Zimta et al, 2020). The changes in gene expression occurred before cells upregulated differentiation-associated genes (Appendix Fig. S8B).

Analysis of the intracellular distribution of SNHG7 showed a predominantly cytosolic location, similar to mRNAs and

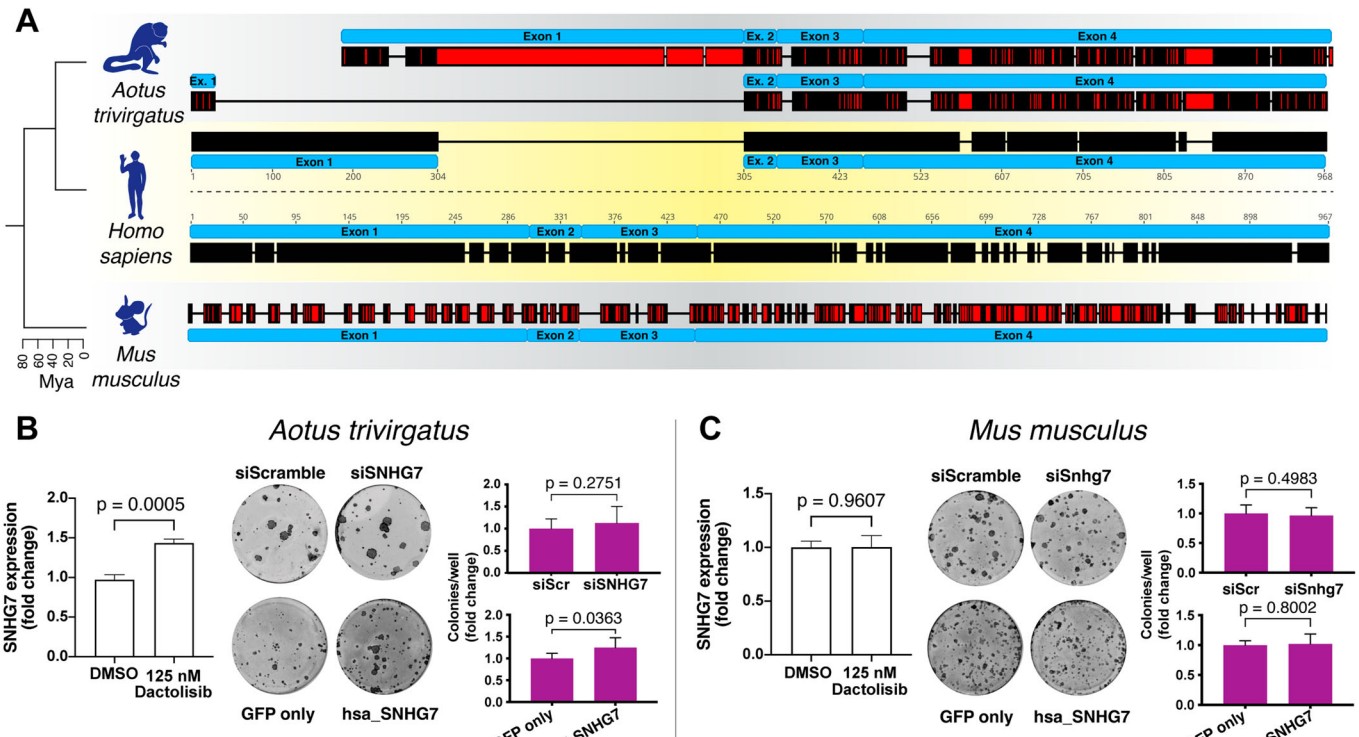

**Figure 4. Human SNHG7 lncRNA can affect cell fate determination in primate species where its activity is lost.**

(A) Alignment of human SNHG7 lncRNA sequence with its orthologs in South American night monkey (*Aotus trivirgatus*, top) and mouse (*Mus musculus*, bottom). A cladogram on the left shows the evolutionary relationship among the species with approximate time separating them. Black colour represents conserved sequence, while red colour represents sequence divergence; thin lines represent gaps in the alignment. (B, C) Regulation and activity of SNHG7 in night monkey (B) and mouse (C) keratinocytes. For each species, shown are expression of SNHG7 after 2 h PI3K treatment (left panels) and clonogenicity assays following either downregulation of endogenous SNHG7 or xenotopic overexpression of human SNHG7 with quantification (right panels). Data shown are mean +/− SD. Unpaired two-tailed t-tests, $n \geq 3$ independent treatments or $n \geq 6$ wells for clonogenicity assays. Source data are available online for this figure.

compatible with the ability to bind miRNAs (Appendix Fig. S8C). To identify which microRNAs could be involved in the biological activity of SNHG7, we uploaded the list of significantly down-regulated ($p_{adj} < 0.05$) genes 24 h after transfection into MIENTURNET (Licursi et al, 2019), a bioinformatic platform designed to infer miRNA regulation based on statistical analysis for over-representation of miRNA-target interactions in a group of genes. We tested for significant enrichment of miRNA targets based on experimentally validated miRNA-target interactions from MiRTarBase (Fig. 5C). The same analysis was also performed for genes significantly downregulated after 48 h (Appendix Fig. S8D). Among the ten most significantly enriched miRNAs, six have the potential to bind to SNHG7. miR-34a-5p and miR-193b-3p have "canonical" miRNA-response elements (MRE) (Bartel, 2009) within the sequence of SNHG7; miR-124-3p can form a non-canonical type of binding that has been previously reported (Agarwal et al, 2015); miR-16-5p, miR-484 and miR-615-3p can potentially form 6-mer "marginal" pairing with SNHG7. Although this latter type of binding is less likely to be conducive to regulation (Agarwal et al, 2015) miR-16-5p targeted the highest number of downregulated genes at both 24 h and 48 h post transfection.

To further investigate the interplay between SNHG7 and miR-16-5p, miR-34-5p, miR-124-3p and miR-193-3p, we performed pathway enrichment analysis of significantly downregulated genes

targeted by each miRNA against the REACTOME database at 24 h post transfection (Fig. 5D). miR-34a-5p and miR-193b-3p targets were predominantly associated with cell cycle and proliferation, while miR-16-5p and miR-124-3p were more associated with signalling and cellular stress.

Since the SNHG7 knockdown phenotype is related to the balance between proliferation and differentiation, we further narrowed down our pool of candidates to miR-34a-5p and miR-193b-3p. The experimentally validated targets of these two miRNAs display a strong bias towards downregulation in the siSNHG7 RNAseq data, in contrast to miR-21-5p, which does not have an MRE within the sequence of SNHG7 and whose targets were not enriched among the significantly downregulated genes (Appendix Fig. S8E). Similarly, looking at the distribution of all predicted targets for these two miRNA families confirmed a significant shift towards downregulation for both miRNAs, which was not observed in the case of miR-21-5p (Fig. S8F). We saw no correlation between the extent of downregulation and the estimated targeting strength of the transcript (Appendix Fig. S8G). SmRNA in situ hybridisation showed that both miRNAs are expressed in keratinocytes (Appendix Fig. S8H). While the activity of the miR-34-5p family of miRNAs in epidermal cells has been described (Antonini et al, 2010), the effect of miR-193-3p on keratinocytes is not known. Consistent with the SNHG7 phenotype, transfection of either

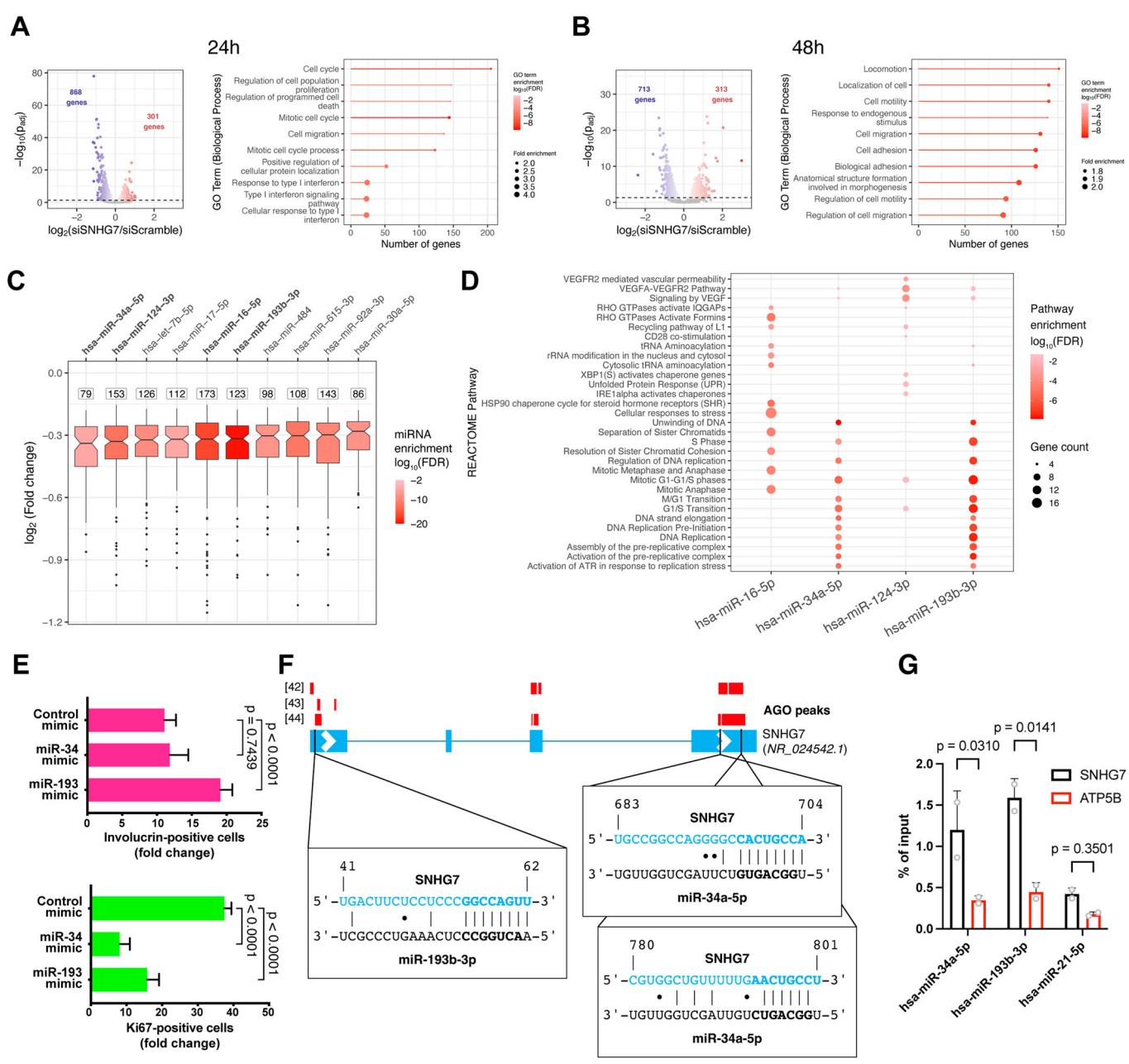

**Figure 5. SNHG7 lncRNA-regulated genes are enriched in targets of SNHG7-interacting miRNAs.**

(A, B) Volcano plot (left) and GO-term enrichment analysis (right) of significantly ($p_{adj} < 0.05$) differentially expressed genes 24 h (A) or 48 h (B) after siSNHG7 transfection. The colour of the lollipops indicates the significance of the GO term enrichment. (C) miRNA response element enrichment in significantly ($p_{adj} < 0.05$) downregulated genes 24 h post-transfection. The number of target genes for each miRNA is shown above the box plot. miRNAs are sorted based on the median downregulation of their target genes. Candidate miRNAs with MREs in the SNHG7 sequence are highlighted in bold. The colour of the boxes indicates the significance of the enrichment. (D) Functional enrichment of the downregulated targets of candidate miRNAs against the REACTOME pathway database. The colour of the points indicates the significance of the enrichment. (E) Effect of miR-34a-5p and mir-193-3p mimics on keratinocyte proliferation and differentiation. (F) Schematic of miRNA binding site location on SNHG7 with location of Ago1/2 peaks from published Clip datasets. Thick lines represent exons, while the thinner lines joining them represent introns. Arrows indicate the direction of transcription. Vertical bars represent Watson-Crick base pairings, while dots highlight G-U wobble base pairings. Seed sequences on miRNAs and MREs on SNHG7 are highlighted in bold. (G) Enrichment of SNHG7 after pulldown of biotinylated candidate miRNAs, a non-biotinylated scrambled control or a biotinylated miRNA without MREs in SNHG7 (miR-21-5p). Enrichment for a control gene without MREs for any of the miRNAs (ATP5B) is also shown. Data shown are mean +/− SD. (E) Ordinary one-way ANOVA and Dunnett's multiple comparison test. $n = 6$ independent transfections. (G) 2-way ANOVA and Fisher's LSD test. $n = 2$ pulldown experiments. Source data are available online for this figure.

miR-34a-5p or miR-193b-3p caused a marked reduction in keratinocyte proliferation, accompanied by a significant increase in differentiation in the case of the miR-193b-3p mimic (Fig. 5E).

In support of the potential for miRNA-based regulation, analysis of published Ago-CLIP datasets (Memczak et al, 2013; Whisnant et al, 2013; Hamilton et al, 2016) showed the presence of peaks of Argonaute binding on the SNHG7 lncRNA transcript overlapping with the MREs for miR-193-3p and miR-34-5p (Fig. 5F). To verify the binding between SNHG7 and the miRNAs, we transfected the cells with biotinylated versions of miR-193b-3p and miR-34a-5p and a control miRNA for which no MRE is present on SNHG7 (miR-21-5p), then performed streptavidin mediated pulldown followed by qPCR. Both miRNAs, but not the negative controls, were able to co-precipitate SNHG7 at higher levels than a negative control gene (ATP5B) and compatible with positive control target genes (Fig. 5G; Appendix Fig. S8I).

We also checked whether the miRNA levels increased after SNHG7 was downregulated, as this could indicate that SNHG7 could operate via Target-Directed miRNA Degradation (TDMD) (Shi et al, 2020; Han et al, 2020). However, we observed no consistent increase of either miR-34a-5p or miR-193b-3p 48 h after transfection (Appendix Fig. S8J).

### Presence of miR-34-5p response elements is necessary for SNHG7-mediated regulation of clonogenicity

Next, we sought to confirm whether miR-34a-5p or miR-193b-3p could contribute to the biological activity of SNHG7. The night monkey SNHG7 does not contain canonical MREs for either of the candidate miRNA families. We thus used night monkey keratinocytes as a null background to assess the activity of human SNHG7 carrying mutations in either the miR-193-3p or miR-34-5p MREs, as well as a control mutation of similar size in the central region of the transcript (Fig. 6A; Appendix Fig. S9A). Stable transfection of miR-193 mutant or control mutant human SNHG7 in night monkey keratinocytes increased their clonogenic potential, similar to the wild-type human transcript. Mutation of the miR-34 MREs abolished the activity of the exogenous transcript (Fig. 6B; Appendix Fig. S9B).

Expression of the wild-type SNHG7 lncRNA was able to rescue the knockdown phenotype in human keratinocytes (Fig. 3I). Conversely, expression of the miR-34-5p MRE mutant was unable to produce any attenuation of the effects of SNHG7 knockdown in human keratinocytes (Fig. 6C–E; Appendix Fig. S9C), further indicating a role for the miR-34-5p MRE in SNHG7 biological activity. We next used a heterologous luciferase reporter system (Salzman et al, 2016) to gauge the effect of SNHG7 expression on individual miR-34-5p targets. Knockdown of SNHG7 caused a significant reduction of the luciferase reporter expression (Fig. 6F). In contrast, overexpression of wild-type SNHG7 lncRNA, but not the miR-34-5p MRE mutant, led to a significant increase of the luciferase reporter (Fig. 6G). The magnitude of this effect was relatively modest, and it should be noted that the reporter assay used a simple short UTR containing a single MRE with very high affinity for miR-34 to guarantee sufficient sensitivity and avoid any potential confounding influence of neighbouring sequences. Endogenous transcripts are much more complex and their susceptibility to miRNA regulation is heavily dependent on both MRE sequence and the UTR context where it is located (for

example, shorter seed pairing could reduce targeting efficiency while the presence of multiple MREs might increase it) (Grimson et al, 2007). Our assay thus simply indicates that SNHG7 activity can significantly affect transcripts responsive to miR-34 regulation. Any direct comparison of the extent of regulation seen in the luciferase assay with that of endogenous transcripts is challenging and should be approached with due caution.

We confirmed that miR-34a-5p is expressed in human epidermis by in situ labelling of healthy skin sections (Appendix Fig. S9D). Despite the lack of activity of human SNHG7 in mouse cells (Fig. 4C), we were also able detect miR-34a-5p expression in both mouse epidermis and cultured keratinocytes (Appendix Fig. S9E,F). Overall, our data point to the involvement of miR-34 response elements in the regulation of keratinocyte clonogenicity mediated by SNHG7.

### Evolutionary conservation of MREs in SNHGs

Our data indicate that SNHG7 regulates clonogenicity in human and night monkey keratinocytes at least partially through a miR-34-5p response element. Therefore, rapidly evolving, short elements in highly expressed non-coding RNA may confer or contribute to the acquisition of new functions over short evolutionary distances.

Short elements such as MREs can potentially arise and disappear rapidly during evolution. In fact, assuming a uniform base frequency and in the absence of selection, a MRE matching any deeply conserved miRNA could potentially occur by chance every ~150 bp. In the case of SNHG7, the functional MRE has emerged recently. If other SNHGs can function in the same way, one would expect to see a range of sequence conservation among MREs. To evaluate this, we used the TargetScan 7 pipeline to identify and score the conservation of MREs in SNHG lncRNA sequences. For the analysis, we only considered broadly conserved miRNA families and only 7-mer and 8-mer MREs. In addition, we excluded SNHGs that had significant overlap with protein-coding genes and transcripts that had exons overlapping the snoRNAs, as this would introduce a conservation bias.

The majority of the sites we identified were shared only among primates. Approximately one quarter were conserved more deeply (Appendix Fig. S9G) and a few passed the TargetScan conservation thresholds (Friedman et al, 2009) (Fig. 6H). A comparison of MRE conservation between the 25 SNHGs included in the analysis and a sample of 250 3′ untranslated regions (UTRs) of coding genes showed similar distributions, albeit with a higher number of deeply conserved sites in UTRs (Appendix Fig. S9H). This increased presence of strongly conserved MREs in UTRs could be reflective of their common susceptibility to miRNA regulation (whereas only a subset of SNHGs might affect miRNA activity), and/or of an increased difficulty in identifying SNHG orthologs in more distant species. Altogether, these data suggest that MREs in SNHGs can in some cases become selected.

## Discussion

In this study, we have identified a group of SNHGs, a class of highly expressed, poorly conserved non-coding genes, that have a pronounced effect on cell fate decisions in human keratinocytes. SNHGs are a known but understudied class of transcripts: they

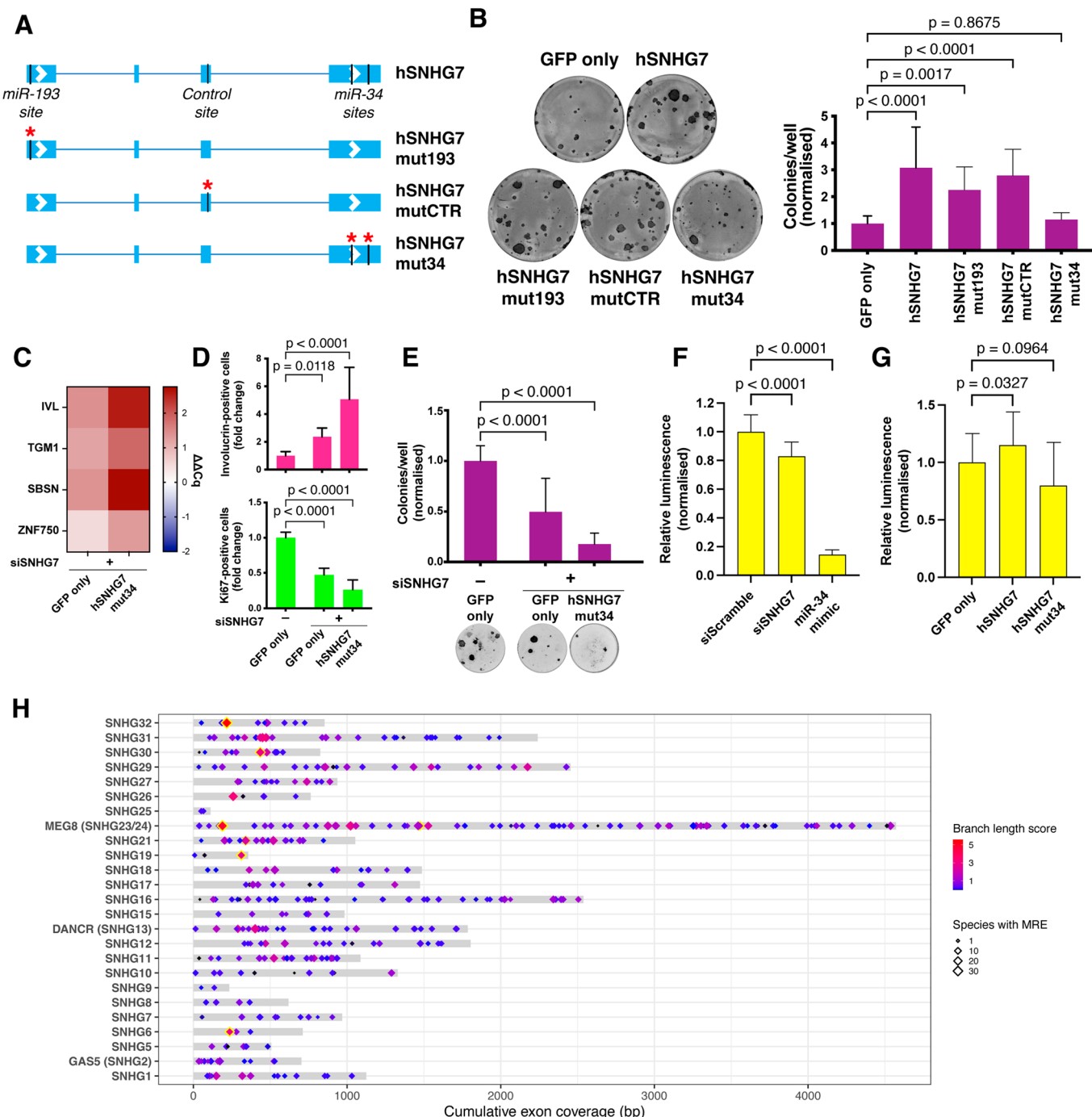

**Figure 6. SNHG7 activity requires the presence of miR-34 binding sites.**

(A) Schematic of SNHG7 mutants. (B) Effect of the xenotopic expression of human SNHG7 mutants on the clonogenicity of night monkey keratinocytes. $n = 12$ wells. (C–E) Rescue of the SNHG7 knockdown phenotype by overexpression of SNHG7 deficient for miR-34 binding sequences. (C) Expression change of multiple differentiation markers in knocked down cells 72 h post-transfection. (D) Staining of differentiation (IVL, top) and proliferation (Ki67, bottom) markers 96 h post-transfection, with quantification. $n \geq 10$ independent transfections. (E) Clonogenicity assay, $n \geq 18$ wells. (F) Effect of SNHG7 knockdown on expression of a heterologous miR-34-sensitive luciferase gene. Transfection of a miR-34 mimic was used as a positive control. $n = 30$ wells. (G) Effect of overexpression of SNHG7 or SNHG7 mutated in its miR-34-5p MREs on the expression of a heterologous miR-34-sensitive luciferase gene. $n \geq 18$ wells. (H) Distribution and conservation of MREs for deeply conserved miRNAs in SNHGs. Yellow borders mark MREs that pass the TargetScan conservation threshold. Data shown in the bar plots are mean $+/-$ SD. (B, D, E) Ordinary one-way ANOVA and Dunnett's multiple comparisons tests. (F, G) Brown-Forsyth and Welch ANOVA and Dunnett's T3 multiple comparison test. Source data are available online for this figure.

have been reported to possess biological activity in cancer cells (Zimta et al, 2020), but only relatively few studies have explored their role in adult tissues (Kino et al, 2010; Kretz et al, 2012). The epidermal expression of SNHGs can be controlled by specific signalling pathways; their levels vary during the early phases of epidermal keratinocyte differentiation, and they are significantly altered in certain skin inflammatory conditions and during wound healing. Knockdown of several of the SNHGs in primary human keratinocytes led to a strong reduction in self-renewal capability. This result is consistent with an early report on the activity of another SNHG (DANCR) in the epidermis (Kretz et al, 2012); however, this gene did not appear in the scRNAseq dataset we analysed. Our data thus suggest that the interplay among multiple signalling pathways affects the levels of several neutrally evolving, non-coding transcripts that influence whether a cell undergoes self-renewal or initiates differentiation and potentially also contribute to stress responses or pathological phenotypes.

In the case of SNHG7, the effect on cell fate determination was acquired recently in the primate lineage. Xenotopic expression of the human transcript could increase the clonogenic capacity of night monkey keratinocytes but did not affect mouse epidermal cells. This differential effect on evolutionarily more distantly related species suggests that SNHG7 activity relies on the presence of context-specific *trans*-acting factors that are present in the monkey cells but absent in the mouse. It is also interesting to note that expression of the human transcript in the monkey cells had a similar effect to overexpression of the transcript in human keratinocytes; however, unlike human, *A. trivirgatus* cells were not dependent on SNHG7 to maintain their ability to self-renew in culture. This indicates that highly expressed, neutrally evolving non-coding transcripts can acquire functionality over short evolutionary distances.

The capacity of SNHG7 to influence keratinocyte fate determination was at least partially dependent on the presence of MREs for a specific miRNA family, miR-34-5p. Knockdown or overexpression of SNHG7 significantly affected miR-34-5p-mediated regulation of an exogenous reporter gene. Regulation of gene expression by competition for the binding of miRNAs has been widely described (Salmena et al, 2011); however, its effectiveness is severely constrained by strict stoichiometric and biochemical parameters (Denzler et al, 2014, 2016; Thomson and Dinger, 2016). Despite this, several instances have been reported where non-coding RNAs of different types can effectively act as sponges for miRNA, mostly in cancer settings (Poliseno et al, 2010; Karreth et al, 2011; Fang et al, 2013) but also in the regulation of neuronal activity (Pollock et al, 2014) and muscle differentiation (Cesana et al, 2011).

Our evaluation of the number of SNHG7 molecules per cell from scRNAseq data, while relatively high, appears incompatible with a mechanism based on direct, indiscriminate competition alone, according to literature estimates (Denzler et al, 2016). Moreover, human SNHG7 was ineffective in regulating cell fate in mouse keratinocytes, even though miR-34-5p is expressed and regulates keratinocyte differentiation in this species (Antonini et al, 2010). These observations suggest that additional factors are likely to be involved in SNHG7 activity. It should also be noted that we cannot exclude the potential contribution of other, miRNA-

independent, molecular mechanisms of action of SNHG7. A full dissection of SNHG7-interacting molecules and detailed investigation of the biochemistry involved in its biological activity will be required to fully understand how it operates.

The expression of intronic snoRNAs, which do not possess their own promoters, is dependent on the transcription of the gene within which they reside (Dieci et al, 2009). This implies that in the case of SNHGs non-coding transcripts retain some selective pressure to be transcribed because of the presence of the snoRNA. The necessity of their transcription for snoRNA production and the lack of strong sequence constraints deriving from the need to code for protein thus potentially optimally places SNHG lncRNAs to function as substrates for the evolution of new functional genetic elements or structures (Palazzo and Koonin, 2020; Stoltzfus, 2012). Indeed, the absence of SNHG7 activity in the epidermal self-renewal/differentiation balance in species where its sequence does not include the MRE for miR-34-5p makes it a paradigmatic example of an "evolutionary spandrel": a transcript whose presence is a by-product of the expression of a separate RNA species (in this case two snoRNAs) and whose sequence is able to neutrally accumulate mutations and act as hotspot for the acquisition of new effects, which can then be subject to the action of drift or selection (Gould and Lewontin, 1979; Stoltzfus, 2012; Doolittle et al, 2014).

The reliance of the biological effect of SNHG lncRNAs on the presence of short MREs, together with their remarkably low sequence conservation, suggests that the fine-tuning of cell fate decisions and potentially other cellular phenotypes could be influenced by the rapid emergence of these genetic elements in abundant transcripts lacking strong constraints on sequence conservation. Due to the short length of MREs, the emergence of sequences with the potential to confer new functions is likely to occur relatively frequently. Most of the MREs, however, would probably be non-functional because (1) not all SNHGs are likely to operate through the same molecular mechanism, and (2) the additional limitations imposed by the context-dependent factors that limit the effectiveness of miRNA sequestration are likely to constrain the intensity and the spread of the effects of SNHGs. We speculate that upon emergence these effects are unlikely to be very strong or ubiquitous, because an excessively penetrant new property would probably be selected against. This is in fact corroborated by the noticeable but relatively limited magnitude of the effect that human SNHG7 expression had in night monkey keratinocytes.

Our data are consistent with a model in which new, limited-scope effects can emerge out of highly expressed, poorly conserved transcripts over brief evolutionary periods and contribute, for example, to the fine-tuning of the balance between cell proliferation and differentiation in adult tissues. Our findings experimentally corroborate the constructive neutral evolution theory and highlight the potential role played by neutrally evolving sequences in the evolution of new genomic functions in the primate lineage (Stoltzfus, 2012; Palazzo and Kejiou, 2022; Palazzo and Koonin, 2020). The extent and the impact of this phenomenon on general organismal diversification and adaptation to changing environments remain to be determined and represent fascinating open questions ripe for further exploration.

# Methods

## Reagents and tools table

| Reagent/Resource | Reference or Source | Identifier or Catalog Number | Notes |
|---|---|---|---|
| **Oligonucleotides and sequence-based reagents** | | | |
| Lincode non-targeting pool | Horizon Discovery | D-001320-10 | |
| Lincode siSNHG7 | Horizon Discovery | R-188323-00 | Individual oligos were used in deconvolution experiments (same catalog, but ending -01 to -04) Oligos -02 and -03 were used to target night monkey SNHG7 due to presence of the same target sequences |
| Lincode siSnhg7 | Horizon Discovery | R-174218-00 | |
| Lincode siSNHG8 | Horizon Discovery | R-188202-00 | |
| Lincode siSNHG12 | Horizon Discovery | R-027562-00 | |
| Lincode siSNHG15 | Horizon Discovery | R-028841-00 | |
| Lincode siSNHG19 | Horizon Discovery | R-189413-00 | |
| miRIDIAN miRNA mimic Negative control | Horizon Discovery | CN-001000-01 | |
| miRIDIAN hsa-miR-34a-5p mimic | Horizon Discovery | C-300551-07 | |
| miRIDIAN hsa-miR-193b-3p mimic | Horizon Discovery | C-300764-05 | |
| **qPCR Primers (Hs:** *Homo sapiens*, **At:** *Aotus trivirgatus*, **Mm:** *Mus Musculus***)** | | | |
| Hs_ATP5B_F | Merck/Sigma | AGGCTGGTTCAGAGGTGTCT | |
| Hs_ATP5B_R | Merck/Sigma | TGGGCAAACGTAGTAGCAGG | |
| Hs_DUSP6_F | Merck/Sigma | ACACAGTGGTGCTCTACGAC | |
| Hs_DUSP6_R | Merck/Sigma | CGGGCTTCATCTTCCAGGTA | |
| Hs_DUSP10_F | Merck/Sigma | GATGCAGCTGAGATTCTGGAC | |
| Hs_DUSP10_R | Merck/Sigma | TAAGTCCACCTCTACCACCAGA | |
| Hs_EVPL_F | Merck/Sigma | TGCAGCACGTGGAGGACTACC | |
| Hs_EVPL_R | Merck/Sigma | CTGTTGCAGCAGCTCTGTGGGG | |
| Hs_HYOU1_F | Merck/Sigma | ACGAGTCCAAGGGCATCAAG | |
| Hs_HYOU1_R | Merck/Sigma | CAGGCTGGAAATGGTGTTGC | |
| Hs_IVL_F | Merck/Sigma | GCCTCAGCCTTACTGTGAGT | |
| Hs_IVL_R | Merck/Sigma | TGTTTCATTTGCTCCTGATGG | |
| Hs_NF2_F | Merck/Sigma | AAAGTGAGGTCTGGCTCTGC | |
| Hs_NF2_R | Merck/Sigma | AGCTGAGATCGTGACACTGC | |
| Hs_PPARA_F | Merck/Sigma | TCGGCGAGGATAGTTCTGGA | |
| Hs_PPARA_R | Merck/Sigma | GGACCACAGGATAAGTCACCG | |
| Hs_PPL_F | Merck/Sigma | AGAGGCGGATGCGGCCATTG | |
| Hs_PPL_R | Merck/Sigma | TTCCCGCGCAGGTTGGTCAC | |
| Hs_RPL13A_F | Merck/Sigma | AACAGCTCATGAGGCTACGG | |
| Hs_RPL13A_R | Merck/Sigma | AACAATGGAGGAAGGGCAGG | |
| Hs_SBSN_F | Merck/Sigma | AAGGCCGGATGCCAGTTTAG | |
| Hs_SBSN_R | Merck/Sigma | TGCTGAATGGCAACCATCAAA | |
| Hs_SNHG7(exon)_F | Merck/Sigma | GACCACGCCTCCCTTTTCAT | |
| Hs_SNHG7(exon)_R | Merck/Sigma | AGGGTCTTAGGTTCCAGGCA | |
| Hs_SNHG7 (intron)_F | Merck/Sigma | TACAGAGGCCAAAGACGCAG | |
| Hs_SNHG7 (intron)_R | Merck/Sigma | GCTGACATTTGGGCCACATC | |
| Hs_SNHG8_F | Merck/Sigma | CGTCTGTGTCATGTGGCCAT | |
| Hs_SNHG8_R | Merck/Sigma | ATCTGGCTCAAACTGACGGT | |
| Hs_SNHG12_F | Merck/Sigma | GCCTCGAGATGGTGGTGAAT | |
| Hs_SNHG12_R | Merck/Sigma | CAGTCCGAAGCGAGAGAAGG | |
| Hs_SNHG15_F | Merck/Sigma | GTTCCTGACAGCCACCTCTC | |
| Hs_SNHG15_R | Merck/Sigma | GCATCTTGGGATTGCTGCTG | |
| Hs_SNHG19_F | Merck/Sigma | TGTGCTACGATCTTGGGACG | |
| Hs_SNHG19_R | Merck/Sigma | AGGGACAAAGTGGTGCGTAG | |
| Hs_SNORA17A_F | Merck/Sigma | GCTGCCGTGTCACATCTGT | |
| Hs_SNORA17A_R | Merck/Sigma | GCTGAGCGTAGACATAGCCT | |
| Hs_SNORA17B_F | Merck/Sigma | TGTTGGCACCACAGACAGTT | |
| Hs_SNORA17B_R | Merck/Sigma | ACTTTCTCTTTGCCCCGAGG | |
| Hs_TBP_F | Merck/Sigma | GTGACCCAGCATCACTGTTTC | |
| Hs_TBP_R | Merck/Sigma | GAGCATCTCCAGCACACTCT | |
| Hs_TGM1_F | Merck/Sigma | GCACCACACAGACGAGTATGA | |
| Hs_TGM1_R | Merck/Sigma | GGTGATGCGATCAGAGGATTC | |
| Hs_ZNF750_F | Merck/Sigma | TTCATGCAGACGAGCCAGAC | |
| Hs_ZNF750_R | Merck/Sigma | ACATTGAGGCTTACTGGAGACT | |
| At_ATP5B_F | Merck/Sigma | TGCTCCCATTCATGCTGAGG | |
| At_ATP5B_R | Merck/Sigma | CCTTGGCATAGGGAGCTAGC | |
| At_RPL13A_F | Merck/Sigma | AAGTACCAGGCAGTGACAGC | |
| At_RPL13A_R | Merck/Sigma | CTCCACGTTCTTCTCGGCTT | |
| At_SNHG7_F | Merck/Sigma | CTCAACTTCCCCTCCTGCAG | |
| At_SNHG7_R | Merck/Sigma | CCGTCACAGGCTAAGTCACC | |
| At_TBP_F | Merck/Sigma | GGAAGAGCAACAAAGGCAGC | |
| At_TBP_R | Merck/Sigma | GCGGTTGTGAGAGTCTGTGA | |
| Mm_Atp5b_F | Merck/Sigma | CTGAGGTCTTCACGGGTCAC | |

| Reagent/Resource | Reference or Source | Identifier or Catalog Number | Notes |
|---|---|---|---|
| Mm_Atp5b_R | Merck/Sigma | GGCTTGTTCTGGGAGATGGT | |
| Mm_Rpl13a_F | Merck/Sigma | CCTGCTGCTCTCAAGGTTGT | |
| Mm_Rpl13a_R | Merck/Sigma | TCCGATAGTGCATCTTGGCC | |
| Mm_Snhg7_F | Merck/Sigma | ATCACGTGCCTGAGGTATGC | |
| Mm_Snhg7_R | Merck/Sigma | AAATTTCCAGCTGCCCAAGG | |
| Mm_Tbp_F | Merck/Sigma | AAGAGAGCCACGGACAACTG | |
| Mm_Tbp_R | Merck/Sigma | TTCACATCACAGCTCCCCAC | |
| At_SNHG7_5RACE_GSP1 | Merck/Sigma | AAATAACTCGGATCTTGGGTTCC | |
| At_SNHG7_5RACE_GSP2 | Merck/Sigma | AGGACTTAAGAACCGGGATACTC | |
| At_SNHG7_5RACE_GSP3 | Merck/Sigma | CAGTGACGTCTCTCTCCTTAATC | |
| At_SNHG7_5RACE_seq | Merck/Sigma | TGTCCTGTAACAGTATTTAGTCCG | |
| At_SNHG7_3RACE_GSP5 | Merck/Sigma | CGGCATCTTCGAGAAATGGATTT | |
| At_SNHG7_3RACE_seq | Merck/Sigma | GGTGTTTCCGTGTGACTGG | |

## scRNAseq analysis

Single-cell RNAseq data from Reynolds et al (2021) was subsetted to include keratinocytes isolated from lesional and non-lesional skin of either atopic dermatitis or psoriasis patients. The two objects thus obtained for each disease state were normalised and scaled using default parameters. The differentially expressed genes between lesional and non-lesional atopic dermatitis and psoriasis skin were identified separately for each condition by running the FindMarkers function with parameters test.use=wilcox and logfc.-threshold=log2(1.5) and keeping all genes with an adjusted *p*-value < 0.01. Analysis of the healthy skin data was described previously (Negri et al, 2023). The RNA molecules per cell were estimated from the scRNAseq data using UMI-based counts mapping to selected genes and then leveraging clustering information to match each cell to its cluster.

## Genome-wide analysis of conservation, expression, and GC content

In order to assess the degree of conservation of different gene classes in the scRNA seq data or genome-wide, the PhastCons scores for all annotated transcripts were calculated by averaging the scores of their exonic portions. Exon coordinates were extracted from the UCSC RefSeq table downloaded from the hg38 human genome assembly on the UCSC Genome browser and used to interrogate the 100-vertebrate (Fig. 1A–C; Appendix Fig. S1A,D) or 30-mammals (Appendix Fig. S1E–G) conservation PhastCons scores (range: 0–1). Transcripts were classified according to the gene_biotype field in the ncbiRefSeqLink table downloaded from the same source. In the genome-wide plot (Fig. 1C; Appendix Fig. S1B), exonic scores were averaged for every transcript. Intron coordinates from 10,000 randomly selected transcripts were used to serve as a "neutrally evolving" control. For the promoter

conservation analysis, the same process was applied to the 500 bp preceding the transcription start site.

Human genome-wide expression scores were obtained from the gtexGeneV8 table downloaded from the hg19 human genome assembly on the UCSC Genome browser. The scores used in Fig. 1E and Appendix Fig. S1B are derived from the total median expression level across all tissues (range: 0–1000). The GTEx data from the same source was also used to generate the tissue-specific expression plots in Appendix Fig. S2. Genes were classified using the geneType field in the table, but some classes were merged as follows: all classes containing the word "pseudogene" were merged into our pseudogene set and classes "lincRNA", "processed_transcript", "antisense", "macro_-lncRNA", "bidirectional_promoter_lncRNA" were merged into our lncRNA set.

Mouse genome-wide expression scores were obtained from the tabulamurisBarChart table downloaded from the mm10 mouse genome assembly on the UCSC Genome browser. The scores used in Appendix Fig. S1C are derived from the total median expression levels across all tissues/cell types (range: 0–1000). Genes were classified according to the gene_biotype field in the ncbiRefSeqLink table downloaded from the same source.

The exon coordinates were also used to interrogate the GC percent content in 5-base windows track (gc5base) downloaded from the hg38 human genome assembly on the UCSC Genome browser.

The rate of evolution of SNHGs was calculated by interrogating the 100-vertebrate conservation PhyloP track from the hg38 human genome assembly on the UCSC Genome browser with the exon coordinates for all SNHG lncRNA transcripts as well as coding gene ACTB as a reference. Since PhyloP scores are basewise $-\log_{10}(p$-values) of conservation (positive scores) or acceleration (negative scores) (Pollard et al, 2010) we used cutoffs of $+2$ for conserved positions or $-2$ for positions undergoing accelerated evolution; any nucleotide with a score between 2 and $-2$ was considered to be evolving neutrally. For each gene, we then calculated the average percentage of nucleotides under the three different evolutionary regimes across all transcripts.

## Primary keratinocyte culture

Primary male human keratinocytes (strain km) isolated from neonatal foreskin or night monkey (*Aotus trivirgatus*) keratinocytes isolated from an oesophageal biopsy (Parenteau et al, 1987) were cultured at 37 °C on mitotically inactivated 3T3-J2 cells in complete FAD medium (Gibco, 041-96624), containing one part Ham's F12, three parts Dulbecco's modified Eagle medium (DMEM) (Gibco, 41966), 100 μM adenine, 10% (v/v) Foetal Bovine Serum (FBS), 1.8 mM $CaCl_2$, 0.5 μg/ml hydrocortisone, 5 μg/ml insulin, 0.1 nM cholera toxin and 10 ng/ml Epidermal Growth Factor (EGF), as described previously (Gandarillas and Watt, 1995). Mouse keratinocytes were isolated from adult back skin and cultured in the same way, except that no $CaCl_2$ was added to the medium (low $Ca^{2+}$ FAD). All animal work was performed under a UK Government Home Office license (PPL Sipila/PP0313918) and approved locally by the Animal Welfare and Ethical Review Body of King's College London (UK). Night monkey keratinocytes were obtained from archival cell stocks originally isolated as described (Parenteau et al, 1987). Primary keratinocytes were used in

experiments at passage 4–7. When subculturing or seeding cells for an experiment, the disaggregated keratinocytes were filtered through a nylon strainer to remove cell clumps and large differentiated cells.

Before mitotic inactivation, 3T3-J2 cells were cultured in DMEM (Gibco, 41966), supplemented with 10% bovine serum.

## Oligonucleotide transfection

Reverse transfection of siRNAs, miRNA mimics or biotinylated miRNAs was performed using INTERFERin transfection reagent (PolyPlus-transfection, 101000016) in accordance with the manufacturer's instructions. All siRNAs and miRNA mimics were purchased from Horizon Discovery. Biotinylated miRNAs were purchased from Integrated DNA Technologies. Oligonucleotides were diluted in OptiMEM medium (Gibco, 31985) and mixed with an appropriate volume of INTERFERin, depending on the transfection vessel. The oligonucleotide/reagent complexes were allowed to form for 15 min at room temperature, before addition of the keratinocytes in Keratinocyte-Serum Free Medium (KSFM) (Gibco, 17005) supplemented with 0.15 ng/ml EGF and 30 mg/ml Bovine Pituitary Extract (BPE). The final concentration of the oligonucleotide, unless otherwise specified, was 30 nM. For transfection of human or monkey keratinocytes tissue culture plastic vessels were coated with 20 µg/ml or 100 µg/ml rat Collagen I (Sigma, C3867) overnight at 4 °C, respectively. For transfection of mouse keratinocytes, tissue culture plastic vessels were coated with extracellular matrix deposited by feeder cells that were removed before keratinocyte addition. Four hours after transfection, the medium was changed to fresh complete KSFM in the case of human or monkey cells or to low $Ca^{2+}$ FAD in the case of mouse cells. For details of the oligonucleotides used see the Reagents and Tools Table.

## Clonogenicity assays

After siRNA transfection or stable lentiviral transduction, 500 (human), 1000 (night monkey) or 5000 (mouse) keratinocytes were plated on a 3T3 feeder layer per well of a six-well dish. After 12 days, feeders were removed, and keratinocyte colonies were fixed in 4% paraformaldehyde (Sigma, 158127) for 10 min then stained with 1% Rhodanile Blue (1:1 mixture of Rhodamine B (Sigma, R6626) and Nile Blue chloride (Sigma, 222550)). The number of colonies was counted manually. Unless otherwise specified, all statistics presented are calculated using well-level data. For siRNA transient transfections, data were collected from at least 3 independent transfections, with the exception of Fig. 2E, which was generated from 2 independent transfections.

## RNA isolation, cDNA synthesis, and quantitative polymerase chain reaction (qPCR)

For mRNA quantification, RNA was extracted using the RNeasy Mini kit (Qiagen, 74104) and subsequently reverse transcribed using the QuantiTect Reverse Transcription (Qiagen, 205311) kit according to the manufacturer's instructions. When mRNA and snoRNA quantification from the same samples was required, RNA was extracted using the miRNeasy Mini kit (Qiagen, 217004) and subsequently reverse transcribed using the mScript II RT kit

(Qiagen, 218161) with HiFlex Buffer according to the manufacturer's instructions. The cDNA thus obtained was diluted to 2.5–5 ng/µl and specific targets were amplified by qPCR using the Fast SYBR® Green Master Mix (Applied Biosystems, 4385612). Expression of all targets was normalised against the expression of three reference genes (RPL13A, ATP5B and TBP) (ΔCq). For miRNA quantification, RNA was extracted using the miRNeasy Mini kit (Qiagen, 217004) and subsequently reverse transcribed using the miRCURY LNA RT kit (Qiagen, 339340) according to the manufacturer's instructions. The cDNA thus obtained was diluted to 2 ng/µl and specific target miRNAs were amplified by qPCR using the miRCURY LNA SYBR Green PCR kit (Qiagen, 339346), with the corresponding primers from the same system. Expression of all targets was normalised against the expression of two reference genes (SNORD48 and U6) (ΔCq). Where indicated, expression was also normalised against control samples (ΔΔCq). For details of the primers used, see the Reagents and Tools Table.

## smRNA (F)ISH and analysis of tissue sections and cultured cells

Chromogenic in situ hybridisation of lncRNA on cultured cells was performed using the RNAscope 2.5 HD Assay – RED kit (ACD, 322350) according to the manufacturer's instructions. Fluorescent in situ hybridisation of lncRNA and mRNA on tissue sections was performed using the RNAscope Multiplex Fluorescent v2 kit (ACD, 323100) according to the manufacturer's instructions. Chromogenic in situ hybridisation of miRNA on cultured cells was performed using the miRNAscope HD Reagent Kit - RED (ACD, 324500) according to the manufacturer's instructions. For tissue staining of mouse and human epidermis with miRNAscope probes, the same protocol was followed, except the development of the FastRED probe was for 20 min and the sections were imaged using the fluorescence of the chromogenic dye, as this detection method offered greater sensitivity. The slides were imaged using a Nikon A1 upright confocal microscope or a Nikon AXR + NSPARC point-scanning confocal microscope.

For quantification, in the case of cultured cells, the number of individual dots in at least 150 cells/staining was counted. In the case of tissue sections, fluorescence intensity was scored by using the nucleus detection algorithm on QPath 0.3.2 and expanding the area around the nucleus by 2 µm. Epidermal layers in Fig. 3D were annotated manually. Positive cell positioning shown in Fig. 3F was quantified by manually thresholding SNHG7 intensity and subsequently scoring the Y position of the positive cells relative to the local minimum and maximum Y coordinates of the epidermis, ranging between 0 (the basal layer) and 1 (the granular layer). The local minima and maxima used in order to account for undulations in the epidermis were calculated as the highest and lowest cell position every 5–20 consecutive cells on the X axis.

## Wound/re-epithelialisation assay of skin biopsies

Surplus surgical waste skin was obtained from consenting patients undergoing plastic surgery. This work was ethically approved by the National Research Ethics Service (UK) (HTA Licence No: 12121, REC No: 14/NS/1073). The tissue was sterilised, washed several times with PBS and cut into 1 cm² pieces. Partial thickness wounds, comprising the epidermis and upper part of the dermis,

were created with a 4 mm punch biopsy. Skin explants were then placed into 6-well hanging cell culture inserts (Millipore) and FAD medium added to create an air–liquid interface. Ex vivo explants were maintained in culture with FAD medium at an air–liquid interface for 2 weeks with media changes every 48 h.

## Immunofluorescence labelling

Plates or slides were washed once with PBS and fixed with 4% Paraformaldehyde (Sigma, 158127) for 10 min at room temperature. Plates or slides were then permeabilised by incubating with 0.2% Triton-X-100 in PBS for 5 min at room temperature, incubated with blocking buffer (10% FBS, 0.25% Fish Skin Gelatin in PBS) for 1 h and stained overnight at +4 °C with the primary antibodies anti-Involucrin (SY3 or SY7 clones), anti-cleaved Caspase 3 (Asp175) (Cell Signalling 9661), anti-Ki67 (Abcam, ab16667), anti-Integrin β1 (eBioscience, 14-0299-82), anti-Keratin 10 (BioLegend, 905404) or anti-Keratin 14 (Biolegend, 906004) diluted to 0.5–1 µg/ml, or according to manufacturer's instructions, in blocking buffer. Plates or slides were then washed three times with PBS, stained with the secondary antibodies AlexaFluor 555 donkey anti-mouse (Invitrogen, A32773), AlexaFluor 647 goat anti-chicken IgY (Invitrogen, A21449) or AlexaFluor 488 donkey anti-rabbit (Invitrogen, A21206) at 1 µg/ml, and the nuclear dye DRAQ5 (abcam, ab108410) at 10 µM or DAPI at 0.1 µg/ml in blocking buffer. Secondary labelling was carried out for 2 h at room temperature in the dark and plates or slides were washed three times with PBS before being imaged using the Perkin-Elmer Operetta High-Content Imaging System, or a Nikon A1R point-scanning confocal microscope.

## High content image analysis

Images acquired with the Perkin-Elmer Operetta High-Content Imaging System were analysed using custom algorithms in the Perkin-Elmer Harmony high-content analysis software package. Nuclei were initially defined using the DRAQ5 or DAPI channel; small (< 50 µm$^2$) and highly irregular (roundness < 0.6) nuclei were excluded from the analysis. To minimise the misattribution of the cytoplasm areas, the level of cytoplasmic staining was inferred from the fluorescence intensity in a ring around the nucleus, as Involucrin staining was homogeneous throughout the cytoplasm. Terminally differentiating cells were identified by manual thresholding of the Involucrin perinuclear fluorescence intensity. The complete Harmony image analysis sequence is available on request.

## OP-Puro assay

General protein synthesis levels were measured using the Click-iT Plus OPP Alexa Fluor 488 Protein Synthesis Assay Kit (Invitrogen, C10456). After 48 h of siRNA transfection, cells were treated with cycloheximide or control diluent for 2 h before O-propargyl-puromycin treatment for 30 min, fixation in 4% paraformaldehyde and staining according to the manufacturer's instructions.

## Skin reconstitution on de-epidermised dermis (DED)

Dermis from excess human adult surgical waste skin was decellularized by repeat freeze/thaw cycles and used as a substrate

for new epidermis formation. Keratinocytes transfected with siRNA targeting SNHG7 or a scrambled control were transferred to decellularized dermal substrates 24 h after transfection and cultured on a tissue culture insert in contact with FAD medium conditioned by a feeder layer at the air–liquid interface for 2 weeks. Dermal substrates were subsequently embedded in optimal cutting temperature compound (OCT, Life Technologies) and frozen before sectioning (12 µm section thickness) and staining with haematoxylin and eosin. Epidermal thickness was scored across the entire length of multiple sections per DED and the mean thicknesses of two DEDs seeded with keratinocytes from two independent transfections per condition were compared.

## Lentiviral transduction of human keratinocytes

Overexpression of wild-type or mutant human SNHG7 in primary human, night monkey or mouse keratinocytes was achieved by lentiviral transduction. Lentiviral particles containing control ("GFP only"), wt human SNHG7 (transcript NR_024542, "hSNHG7") or mutant human SNHG7 lncRNAs ("hSNHG7mut", see Fig. 6A and Appendix Fig. S9A) custom overexpression plasmids (Oxgene) were transduced into keratinocytes for 24 h at a multiplicity of infection of 3 m.o.i. in the presence of 5 µg/ml polybrene. Both control and overexpression plasmids encoded GFP as an indicator of transduction efficiency. Stably transduced cells were FACS-sorted 48 h after transduction and the GFP-positive cells were subcultured and expanded. GFP transcription was under the control of a separate promoter to avoid the generation of a hybrid RNA sequence that may have interfered with SNHG7 function. Plasmid sequences are available on request.

## Suspension-induced keratinocyte differentiation

Keratinocytes were differentiated in suspension as described (Adams and Watt, 1989; Mishra et al, 2017). Pre-confluent cultures were disaggregated in trypsin/EDTA and resuspended at a concentration of 10$^5$ cells/ml in medium containing 1.45% methylcellulose. Aliquots were plated in 6-well plates coated with 0.4% polyHEMA; this ensured that there was no cell-substratum adhesion. The suspended cells were subsequently incubated at 37 °C. At each collection time point, the methylcellulose was diluted with PBS and the cells recovered by centrifugation.

## 3′ and 5′ RACE

Sequencing of *Aotus trivirgatus* SNHG7 lncRNA was achieved by amplifying the 3′ and 5′ end of the cDNA from internal primers using the 5′/3′ RACE kit, 2nd Generation (Roche, 03353621001). Since no genome data was available for *Aotus trivirgatus*, internal primers were designed from the sequence of the closely related species *Aotus nancymaae*. For primer details, see the Reagents and Tools Table.

## Sequence alignment

*A. trivirgatus* SNHG7 lncRNA sequences were aligned to the human transcript (NR_024542) by using a supervised application of the Needleman-Wunsch global alignment algorithm on an exon-by-exon basis. The intron-exon structure of the *A. trivirgatus* gene

was inferred by manually comparing the genomic sequences of *H. sapiens* and *A. nancymaae* (NW_018496780) for which an assembled genome was available.

The *M. musculus* SNHG7 sequence (NR_024068) was aligned to the human transcript using a supervised application of the Needleman-Wunsch global alignment algorithm on an exon-by-exon basis using the annotated exons.

Alignments were visualised using the Geneious Prime software (Geneious).

## RNAseq analysis

Samples transfected with siRNA targeting SNHG7 or a scrambled control were harvested 24 h and 48 h post-transfection and sent to Azenta Life Sciences for bulk RNA sequencing. Sequence reads were trimmed to remove possible adaptor sequences and nucleotides with poor quality using Trimmomatic v.0.36. The trimmed reads were mapped to the Homo sapiens GRCh38 reference genome available on ENSEMBL using the STAR aligner v.2.5.2b. Unique gene hit counts were calculated by using featureCounts from the Subread package v.1.5.2. Only unique reads that fell within exon regions were counted.

After extraction of gene hit counts, the gene hit counts table was used for downstream differential expression analysis. Using DESeq2, a comparison of gene expression between the knockdown (siSNHG7) and control (siScramble) samples was performed at 24 h and 48 h post transfection. The Wald test was used to generate *p*-values and $\log_2$ fold changes. The *p*-values were then adjusted for multiple testing using the Benjamini and Hochberg method. The adjusted *p*-values for this test are referred to in the manuscript as $p_{adj}$.

GO term enrichment analysis was performed using ShinyGO 0.77 (Ge et al, 2019), inputting all significantly differentially expressed genes ($p_{adj} < 0.05$) at each time point and assessing enrichment of Biological Process GO terms with respect to all genes detected in the RNA-seq.

Significantly ($p_{adj} < 0.05$) downregulated genes at each time point were scanned for enrichment of miRNA response elements (MREs) against the miRTarBase database (version 7.0) of experimentally validated miRNA-mRNA interactions using MIEN-TURNET (Licursi et al, 2019). Briefly, this web tool performs a hypergeometric test to assess the probability of finding a certain minimum number of interactions (or more) between a miRNA and a list of target genes, considering the total number of interactions that miRNA engages in, and the total number of miRNA-target interactions present in the database, according to the formula:

$$p = 1 - \sum_{i=0}^{X-1} \frac{\binom{K}{i}\binom{M-K}{N-i}}{\binom{M}{N}}$$

*M* is the universe of total interactions encompassed in the miRTarBase database, *K* is the total number of interactions miRNA *i* engages in, *N* is the number of genes in the candidate list (in our case the significantly downregulated genes following SNHG7 knockdown), and *X* is the number of miRNA interactions found within the candidate list. The *p*-values thus calculated are subsequently adjusted for multiple testing using the Benjamini–Hochberg method. The adjusted *p*-values for this test are referred to as FDR in the manuscript.

The same tool was used to assess enrichment of REACTOME Pathways for the downregulated targets of each candidate SNHG7-interacting miRNA.

## Subcellular fractionation

Isolation of cytoplasmic and nuclear fractions from human keratinocytes was performed by following a published protocol (Gagnon et al, 2014). After fractionation and RNA extraction, all samples were diluted to the same RNA concentration and 250 ng of RNA from each sample spiked with 100 ng mouse RNA to control reaction efficiency were reverse transcribed using the miScript II RT kit (Qiagen, 218161) with HiFlex Buffer according to the manufacturer's instructions. Amounts in each fraction were then calculated by multiplying the RNA found in 250 ng by the total volume of each fraction.

## RNA pulldown

Pulldown of biotinylated microRNAs was performed as described in the literature (Michelini et al, 2017) 24 h after transfection with biotinylated miRNAs purchased from Integrated DNA technologies. Before pulldown, 1/10th of each sample was aliquoted to serve as the input reference and the RNA pulled down in each sample was normalised to the corresponding input.

## Heterologous luciferase assay

To detect miR-34 activity, we used a dual luciferase reporter plasmid, psiCHECK2-miR-34 WT (Salzman et al, 2016) (Addgene plasmid # 78258; http://n2t.net/addgene:78258; RRID:Addgene_78258). The plasmid contains two luciferase genes: the firefly luciferase is independent of miR-34 activity and is used as a transfection efficiency control; the *Renilla* luciferase gene contains a miR-34-5p response element in its 3′UTR.

In one set of experiments, primary human keratinocytes stably overexpressing wt SNHG7, SNHG7 carrying mutations in the miR-34-5p binding sites, or GFP only were transfected with psiCHECK2-miR-34 WT using JetPRIME transfection reagent (PolyPlus-transfection, 101000001), as per the manufacturer's instructions. Alternatively, primary human keratinocytes were co-transfected with psiCHECK2-miR-34 WT and a miR-34a-5p mimic, or a siRNA pool targeting SNHG7, or a control scramble siRNA, using the same transfection strategy. Transfections were carried out in 96-well plates using 0.3 μg plasmid DNA/well either alone or with 30 nM siRNA, and the plasmid or plasmid/oligo mixes were incubated with 0.6 ml/well of JetPRIME reagent. The nucleic acid/reagent complexes were allowed to form in JetPRIME Buffer for 10 min at room temperature before addition of the keratinocytes in Keratinocyte-Serum Free Medium (KSFM) (Gibco, 17005) supplemented with 0.15 ng/ml EGF and 30 mg/ml Bovine Pituitary Extract (BPE). The medium was changed to fresh KSFM 4 h after transfection. Two days after transfection, the cells were assayed with the DualGlo Luciferase Assay System (Promega E2920) according to the manufacturer's instructions. Wells where the firefly luciferase intensity was less than double the blank wells were excluded from the analyses. The *Renilla* intensity in each well was normalised to the firefly luciferase expression to correct for variations in transfection efficiency. Statistics were calculated using well-level data generated from at least 3 independent cultures, with

the exception of the SNHG7mut34-overexpressing line for which data were generated from 2 independent cultures.

## MRE conservation analysis

In order to assess the presence and conservation of miRNA response elements (MREs) in SNHGs we used the "maximum exonic coverage" of all transcripts that did not overlap coding sequences or snoRNAs for each of the SNHGs.

First, SNHG transcripts that did not contain exons overlapping coding genes or snoRNAs were selected and the coordinates for their exons were extracted from the hg19 human genome assembly on the UCSC Genome browser. Second, the exonic coordinates were used to extract the sequences from the 100-vertebrates multiple sequence alignment (multiz100way) from the hg19 human genome assembly on the UCSC Genome browser. Third, the extracted sequences were stitched together using the minimum and maximum coordinates of all selected transcripts for each SNHG. This generated, for every SNHG, a multiple sequence alignment of all exonic portions of the gene stitched together in an idealised "maximum exonic coverage" transcript. Fourth, we used the TargetScan 7 suite to identify MREs in the species present in the multiple sequence alignment and to calculate their branch length scores. For this analysis, we did not require miRNAs to have been annotated in each species but only seed sequences of deeply conserved miRNAs were used. Since not all individual isoforms are analysed, it is possible that some additional miRNA sites can be created at alternatively spliced junctions.

In order to compare the distributions of MRE conservation between SNHGs and the 3′UTRs of coding genes, we performed the same MRE conservation analysis on 250 UTRs maintaining the same background conservation binning (Friedman et al, 2009) distribution we had for the SNHG set.

Some of the steps in this analysis were performed using the Galaxy platform (Afgan et al, 2022).

## Statistical information

R and GraphPad Prism (Version 10) were used for statistical analysis. Sample normality was assessed by Shapiro–Wilks testing and/or visual inspection of Q–Q plots. In general, for normally distributed samples, two-tailed unpaired t-tests were used when only two groups were compared, while ANOVA was employed when three or more groups were analysed, followed by multiple comparison tests. For samples where normality could not be assumed, Kolmogorov–Smirnov tests were used when only two groups were compared, while the Kruskal–Wallis test was employed when three or more groups were analysed, followed by multiple comparison tests. Details of the test performed in individual experiments are provided in the figure legends. For statistical tests used in the analysis of scRNAseq and RNAseq data, see the respective Methods section. Samples were excluded from analysis only in the case of ineffectual experimental treatment (e.g. lack of expression reduction in knockdown experiments) or abnormalities (e.g. individual well contaminations).

## Data availability

The data for the RNAseq experiment reported in this study has been deposited into the Gene Expression Omnibus (GEO), with accession number GSE263910 (https://www.ncbi.nlm.nih.gov/geo/query/acc.cgi?acc=GSE263910).

The source data of this paper are collected in the following database record: biostudies:S-SCDT-10_1038-S44318-024-00172-8.

## Peer review information

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

## Acknowledgements

MVR would like to thank Dr Ajay Mishra for early insights about this project and Dr Flavia Michelini for useful advice about biotin-mediated RNA pulldown experiments. FMW gratefully acknowledges financial support from Cancer Research UK (C219/A23522), the Medical Research Council (MR/PO18823/1) and the Wellcome Trust (206439/Z/17/Z; 211276/E/18/Z). We are also grateful to the National Institute for Health Research (NIHR) Biomedical Research Centre based at Guy's and St Thomas' NHS Foundation Trust and King's College London. The views expressed are those of the author(s) and not necessarily those of the NHS, the NIHR or the Department of Health.

## Author contributions

**Matteo Vietri Rudan**: Conceptualization; Data curation; Formal analysis; Validation; Investigation; Visualization; Writing—original draft; Writing—review and editing. **Kalle H Sipilä**: Conceptualization; Investigation; Writing—review and editing. **Christina Philippeos**: Investigation. **Clarisse Ganier**: Formal analysis; Investigation. **Priyanka G Bhosale**: Investigation. **Victor A Negri**: Formal analysis. **Fiona M Watt**: Conceptualization; Supervision; Funding acquisition; Writing—review and editing.

Source data underlying figure panels in this paper may have individual authorship assigned. Where available, figure panel/source data authorship is listed in the following database record: biostudies:S-SCDT-10_1038-S44318-024-00172-8.

## Disclosure and competing interests statement

The authors declare no competing interests. Fiona Watt is the director of EMBO. *The EMBO Journal* is editorially independent of EMBO.

