## [Peer Review File · The EMBO Journal]

Neutral evolution of snoRNA Host Gene long non-coding RNA affects cell fate control

Matteo Vietri Rudan, Kalle Sipilä, Christina Philippeos, Clarisse Gânier, Priyanka Bhosale, Victor Negri, and Fiona Watt

Corresponding author: Fiona Watt (fiona.watt@embo.org)

Review Timeline:

Transferred from Review Commons:	23rd Feb 24
Editorial Decision:	8th Mar 24
Revision Received:	11th Apr 24
Editorial Decision:	4th Jun 24
Revision Received:	15th Jun 24
Accepted:	1st Jul 24

Editor: Daniel Klimmeck

Transaction Report:

This manuscript was transferred to The EMBO Journal following peer review at Review Commons.

Review #1

1. Evidence, reproducibility and clarity:

Evidence, reproducibility and clarity (Required)

Vietri Rudan and colleagues present an interesting study to examine the function and mechanism of small nucleolar RNA Host Gene 7 (SNHG7) in human keratinocytes. The key findings are that a widely expressed SNHG7, a long noncoding RNA hosting two snoRNAs in the introns, promotes keratinocyte proliferation and inhibits differentiation likely by sequestering miR34a from its targets. The experiments are generally well-executed with proper controls. Notably, they leveraged human, night monkey and mouse keratinocytes to reveal primate specific functions of SNHG7 in a miR34 dependent manner. I have a few comments and suggestions that should be addressed to further strengthen the study.

1. Single-cell RNAseq (Fig. 1B) and in situ (Fig. 3D) results both indicate that SNHG7 is broadly expressed in multiple epidermal layers but more enriched in the spinous layer. Although most assays, such as colony formation and Ki67 staining, did not specifically examine the role of SNHG7 in the spinous layer, the raft culture experiment seemed to indicate specific reduction of the spinous layer (Fig. 3H), which was more prominent than basal defects. The authors should examine the defects more carefully in the raft culture system by using basal, spinous and granular markers. It is possible that SNHG7 functions to maintain limited cell proliferation while restrict premature differentiation. In addition, they should perform serial passage experiments to distinguish whether overexpression of SNGH7 can indeed confer self-renewal in long-term experiments. Based on these results, they may need to refine their hypothesis/conclusion whether SNHG7 functions primarily on stem cell self-renewal or transiently maintain proliferation in transitioning cells.

2. The main proposed mechanism is the sequestration of miR34 by SNHG7. While miR34 is well known for its function in inhibiting cell proliferation, the ability of coding or noncoding RNA to sequester miRNAs is highly dependent on the stability and copy number of these RNAs. Since they have single-cell data with UMI information, they should estimate the copy number of SNHG7 in epithelial cell populations, and this could provide a range for the "buffering" capacity of SNHG7. They should also examine, ideally by in situ hybridization, the expression patterns of miR34 in human vs mouse skin. While miR34 expression can be induced by p53 activation, it is possible that its expression varies in different species. It'll be interesting to determine whether the lack of miR34 expression in mouse keratinocyte or mouse skin could explain the insensitivity of mouse keratinocytes to SNHG7. Finally, to further demonstrate the competition between SNHG7 and miR34 targets, they can use a heterologous luciferase reporter system with a canonical miR34 targeting site in the 3'UTR and quantify luciferase activities with or without SNHG7 (or SNHG7 mut34 variant). This assay could quantify the impact of SNHG7 on individual miR34 targets.

2. Significance:

Significance (Required)

This study reveals a potential function and mechanism of primate specific noncoding RNA for its role in modulating gene expression and cellular functions in the skin. It provides a new paradigm for identifying molecular functions of poorly conserved RNAs.

3. How much time do you estimate the authors will need to complete the suggested revisions:

Estimated time to Complete Revisions (Required)

(Decision Recommendation)

Less than 1 month

No

Review #2

1. Evidence, reproducibility and clarity:

Evidence, reproducibility and clarity (Required)

****Summary****

The manuscript submitted by M. Vietri Rudan and co-authors presents a functional analysis of a specific class of long non-coding RNA transcripts, namely the small nucleolar RNA host genes (SNHG). These genes are defined by the fact that snoRNA molecules are embedded within their loci, often in the SNHG introns. The rationale presented by the authors for studying this specific class of lncRNA genes is the fact that they display very low levels of evolutionary sequence

conservation (even compared to the generally low conserved lncRNA transcripts) and that they are expressed at remarkably high levels (contrary to most lncRNA genes, which are very weakly expressed). The overarching goal of this study thus appears to be determining whether functional transcripts can exist in the absence of evolutionary conservation.

The authors study SNHG7 in the human keratinocytes model system. They analyze the expression levels of SNHG7 in skin affected by atopic dermatitis and psoriasis, compared to normal skin. They identify several SNHG7s that are differentially expressed between diseased and normal skin, and which are also detected at strong expression levels in single cell RNA-seq assays of human keratinocytes. The authors use knockdown assays to investigate the potential roles of these SNHG7s in keratinocytes and show that the knockdown of each of 5 selected SNHG7s results in a loss of clonogenicity.

The last part of the manuscript focuses on the functional characterization of SNHG7, chosen because it is dysregulated in both atopic dermatitis and psoriasis and because its knockdown strongly affects clonogenicity. Additional assays showed that SNHG7 knockdown results in a reduced rate of proliferation and an increase in the fraction of differentiating cells. To ensure that the effects of the knockdown are not simply due to the absence of the snoRNA molecules embedded in the SNHG7 locus, the authors overexpressed the spliced form of SNHG7 (which lacks the snoRNA genes), successfully rescuing the cellular phenotype. They also verified that the knockdown does not affect the abundance of the corresponding snoRNA molecules.

To propose a potential mechanism for the involvement of SNHG7 in keratinocyte proliferation, the authors investigated its capacity to act as a miRNA "decoy". They searched for an enrichment of miRNA binding sites among the genes that are downregulated following SNHG7. They identified several miRNAs which are predicted to target SNHG7 as well as (a substantial fraction of) the genes that are downregulated following SNHG7 knockdown. The transfection of two of these miRNAs (miR-193-3p and miR-34-5p) has a negative effect on keratinocyte proliferation, supporting the authors's hypothesis that SNHG7 may act via "sponging" these miRNAs, thereby positively contributing to the control of the expression of the other miRNA target genes. Consistent with this hypothesis, the authors show that overexpression of a SNHG7 mutant sequence that lacks the miRNA binding sites is not able to rescue the knockdown phenotype.

However, this hypothesis is not supported (or at least not further reinforced) by the evolutionary analysis performed by the authors. They were able to identify homologues for SNHG7 in another primate species (night monkey) and in the mouse. Transfection of the human SNHG7 sequence was able to increase clonogenicity in night monkey cells, but not in mouse cells. Given that miR-34 is also present in mouse and was previously shown to affect keratinocyte proliferation, it is not clear why the human SNHG7 sequence is not able to act as a miRNA sponge in this species. Likewise, only a slight effect on clonogenicity is observed upon SNHG7 transfection in night monkey. The authors conclude (correctly, in my view) that further investigations are needed to confirm the potential functions of SNHG7.

Overall, I find that this study is interesting and carefully conducted. Nevertheless, I have several comments that I hope can improve this manuscript.

****Major comments:****

1. The premise of the study relies on the observation that SNHG7s have low levels of sequence conservation. Indeed the authors aim to prove that biological functions can be identified even in the absence of evolutionary conservation. However, the extent of evolutionary conservation strongly depends on the phylogenetic scale at which it is analyzed. Here, the authors evaluate sequence conservation using PhastCons and PhyloP scores determined using alignments of human and 99 other vertebrate species. These scores reflect the extent of long-term sequence conservation. At this scale, only a small percentage of the human genome can be considered to be "conserved". It is thus not surprising that SNHG7 and other lncRNAs are not conserved at this scale, even if they carry some biological functions in the human genome. Here, it would be useful to redo the sequence conservation analyses using PhastCons and PhyloP scores computed on less distant species. Pre-computed scores exist for alignments containing human and other mammalian species, mainly primates (UCSC genome browser). It would also be good to provide comparisons of sequence conservation levels on the snoRNA genes and on the non-snoRNA parts of SNHG7s. In addition to protein-coding genes, pseudogenes and lncRNAs, it would be good to add a perfectly neutral control for the sequence conservation in Figure 1C - for example, flanking intergenic regions for SNHG7s. It might also be a good idea to analyze the GC content of SNHG7s compared to other lncRNAs, since GC content can be correlated with sequence conservation levels, in particular in noncoding regions.

Importantly, the SNHG7 selected for detailed investigation (SNHG7) appears to be more conserved than the bulk of human lncRNAs, given that it is found in another primate and in mouse. It would be interesting to analyze in details what sequence features of this lncRNAs are conserved among species - for example, are the SNHG7 splice sites and promoter regions conserved? Are the snoRNA genes always located in the introns?

2. I am not perfectly convinced by the enrichment of miRNA target genes among the genes that are downregulated upon SNHG7 knockdown. The methods do not clearly explain how this enrichment is calculated. What is the background used for this enrichment analysis? In Figure 5C, we see that the genes predicted to be targeted by the top 10 microRNAs tend to have negative fold changes in the differential expression analysis (downregulation following knockdown). However, from Figure 5A it seems that the great majority of significantly differentially expressed (DE) genes have negative fold changes. How do the miRNA target genes differ from all other DE genes? What proportions of all predicted miRNA target genes (expressed in keratinocytes) are DE following knockdown, and how does this compare with the target genes of other miRNAs?

3. If keratinocyte RNA-seq data is available for other species (for example mouse), it would be interesting to test whether the high expression levels of SNHG7 and the other analyzed SNHG7s are also conserved in the other species.

****Minor comments:****

1. The AD (atopic dermatitis) abbreviation should be explained the first time it is mentioned in the text.
2. More details are needed regarding the MIENTURNET analyses in the methods and in the main text
3. Figure 5C, it is not clear how to interpret the color code for the boxplot. Does this represent the median or mean FDR of the target genes? Are only genes with FDR<5% included in this analysis?
4. Figure 6D, I am not sure how to the D panel. Do the gray rectangles represent the exonic length of the SNHGs? Do the dots correspond to the positions of the miRNA target sites? Here, a more quantitative comparison with the extent of sequence conservation of miRNA binding sites in SNHG, other lncRNAs and in protein-coding genes would be perhaps better suited.

2. Significance:

Significance (Required)

Overall, this study is highly relevant in the field of lncRNA functionality and evolution. It presents evidence for a potential involvement in the regulation of cell proliferation for SNHG, a role that appears to be independent of the snoRNAs produced by these loci. This study reinforces the current body of work suggesting that lncRNAs and other noncoding transcripts can sometimes function as miRNA "decoy" targets. This study will be of interest for a specialized audience, oriented towards understanding lncRNA biological functions.

3. How much time do you estimate the authors will need to complete the suggested revisions:

Estimated time to Complete Revisions (Required)

(Decision Recommendation)

Between 1 and 3 months

Yes

Revision Plan

Manuscript number: RC-2023-02280

Corresponding author(s): Fiona Mary, Watt

[The "revision plan" should delineate the revisions that authors intend to carry out in response to the points raised by the referees. It also provides the authors with the opportunity to explain their view of the paper and of the referee reports.]

The document is important for the editors of affiliate journals when they make a first decision on the transferred manuscript. It will also be useful to readers of the reprint and help them to obtain a balanced view of the paper.

*If you wish to submit a full revision, please use our "Full Revision" template. **It is important to use the appropriate template to clearly inform the editors of your intentions.**]*

1. General Statements [optional]

We would like to sincerely thank both reviewers for taking the time to examine our work, for the cogent points they have raised, and their constructive attitude aimed at improving our manuscript.

2. Description of the planned revisions

Reviewer #1

2. The main proposed mechanism is the sequestration of miR34 by SNHG7. While miR34 is well known for its function in inhibiting cell proliferation, the ability of coding or noncoding RNA to sequester miRNAs is highly dependent on the stability and copy number of these RNAs. Since they have single-cell data with UMI information, they should estimate the copy number of SNHG7 in epithelial cell populations, and this could provide a range for the "buffering" capacity of SNHG7. They should also examine, ideally by in situ hybridization, the expression patterns of miR34 in human vs mouse skin. While miR34 expression can be induced by p53 activation, it is possible that its expression varies in different species. It'll be interesting to determine whether the lack of miR34 expression in mouse keratinocyte or mouse skin could explain the insensitivity of mouse keratinocytes to SNHG7. Finally, to further demonstrate the competition between SNHG7 and miR34 targets, they can use a heterologous luciferase reporter system with a canonical miR34 targeting site in the 3'UTR and quantify luciferase activities with or without SNHG7 (or SNHG7 mut34 variant). This assay could quantify the impact of SNHG7 on individual miR34 targets.

Revision Plan

We will include the analysis of the scRNAseq data to estimate the copy number of SNHG7 in the epidermal populations.

We will also perform in situ hybridization staining for miR-34 in human and mouse epidermis as well as mouse keratinocytes.

Finally, we will carry out the luciferase reporter experiments.

Reviewer #2

3) If keratinocyte RNA-seq data is available for other species (for example mouse), it would be interesting to test whether the high expression levels of SNHG7 and the other analyzed SNHG7s are also conserved in the other species.

We will include the RNA-seq data, if available.

3. Description of the revisions that have already been incorporated in the transferred manuscript

Reviewer #1

1. Single-cell RNAseq (Fig. 1B) and in situ (Fig. 3D) results both indicate that SNHG7 is broadly expressed in multiple epidermal layers but more enriched in the spinous layer. Although most assays, such as colony formation and Ki67 staining, did not specifically examine the role of SNHG7 in the spinous layer, the raft culture experiment seemed to indicate specific reduction of the spinous layer (Fig. 3H), which was more prominent than basal defects. The authors should examine the defects more carefully in the raft culture system by using basal, spinous and granular markers. It is possible that SNHG7 functions to maintain limited cell proliferation while restrict premature differentiation. In addition, they should perform serial passage experiments to distinguish whether overexpression of SNHG7 can indeed confer self-renewal in long-term experiments.

We have included staining of a range of epidermal markers in the raft cultures (ITGB1, K14, K10 and IVL) in the revised manuscript (Fig. S6B). We do not observe major changes in the distribution of any of the differentiation markers.

In terms of serial passage, we have cultured SNHG7-overexpressing or control cells for multiple passages until their growth capacity was exhausted. The SNHG7-overexpressing cells grew for approximately seven more passages than the control cells. We have added this information to the revised manuscript (Fig. S6L).

Reviewer #2

1) The premise of the study relies on the observation that SNHG7s have low levels of sequence conservation. Indeed the authors aim to prove that biological functions can be identified even in the absence of evolutionary conservation. However, the extent of evolutionary conservation strongly depends on the phylogenetic scale at which it is analyzed. Here, the authors evaluate sequence conservation using PhastCons and PhyloP scores determined using alignments of

Revision Plan

human and 99 other vertebrate species. These scores reflect the extent of long-term sequence conservation. At this scale, only a small percentage of the human genome can be considered to be "conserved". It is thus not surprising that SNHG and other lncRNAs are not conserved at this scale, even if they carry some biological functions in the human genome. Here, it would be useful to redo the sequence conservation analyses using PhastCons and PhyloP scores computed on less distant species. Pre-computed scores exist for alignments containing human and other mammalian species, mainly primates (UCSC genome browser). It would also be good to provide comparisons of sequence conservation levels on the snoRNA genes and on the non-snoRNA parts of SNHGs. In addition to protein-coding genes, pseudogenes and lncRNAs, it would be good to add a perfectly neutral control for the sequence conservation in Figure 1C - for example, flanking intergenic regions for SNHGs. It might also be a good idea to analyze the GC content of SNHGs compared to other lncRNAs, since GC content can be correlated with sequence conservation levels, in particular in noncoding regions. Importantly, the SNHG selected for detailed investigation (SNHG7) appears to be more conserved than the bulk of human lncRNAs, given that it is found in another primate and in mouse. It would be interesting to analyze in details what sequence features of this lncRNAs are conserved among species - for example, are the SNHG7 splice sites and promoter regions conserved? Are the snoRNA genes always located in the introns?

The reviewer raises very good points. We have added the conservation data of snoRNA genes (all or SNHG-resident, both significantly more conserved than SNHGs or lncRNA), and a "true neutral" control (we used the introns of 10,000 randomly sampled genes) to our PhastCons analysis (Fig.1C). We also added a conservation analysis of the promoters (defined as the 500 bp upstream of the transcription start site) of coding genes, pseudogenes, lncRNA and SNHGs (Fig, S1B). We have now performed all our conservation analyses using both PhastCons scores generated from the 100-vertebrate alignment and PhastCons scores generated from the 30-mammals (28 primate) alignment (Fig. S1E-G). We do not detect any marked difference between the two alignment sets. We have also added a comparison of the GC content in lncRNA, SNHGs, coding genes and introns (Fig.S1C).

Due to the nature of the PhyloP scores, the 30-mammal PhyloP track (phyloP30) would be unsuitable to detect additional conservation in the primate lineage using the thresholding analysis we employed in Fig. 1E. The PhyloP track gives base-wise p-values for conservation (positive values) or accelerated evolution (negative values). Using alignments of genomes that are overall more similar to each other (as in the 30-mammal alignment set) makes it more difficult to distinguish between conserved and neutrally evolving regions, because even segments that are not under constraint will look relatively similar due to the evolutionary proximity of the species in the set. For the same reason this alignment set is quite sensitive to accelerated evolution, as it contains many relatively similar genomes in the alignment.

This causes the PhyloP30 scores to be very asymmetrical around zero: the conservation (positive) scores never reach 2 (p-value of 0.01) in the whole track (not even coding regions of very well-conserved genes), while acceleration scores can reach very significant values, down to -20. Conversely, the PhyloP100 track (used in Fig.1E) is quite symmetrical around 0 and is thus better suited for the purposes of the analysis in Fig.1E, which are to detect both conserved and accelerated portions of SNHGs. We have however also inspected the PhyloP30 track manually and do not observe any clear evidence of presence of additional conserved elements in SNHG7. We have added all conservation tracks for SNHG7 to Fig. 3A.

While lncRNA orthologs can be identified by using a combination of sequence conservation, conserved synteny with surrounding genes and in some cases conserved gene structure, SNHG orthologs can additionally be identified by the embedded (conserved) intronic snoRNA sequence, which makes them easier to find even when the transcript sequence bears no

Revision Plan

similarity across species. The mouse SNHG7 sequence, for example, does not match with human SNHG7 even using the least stringent BLAST parameters. The monkey sequence is similar enough to match with the human in the 3' of the gene, but the intron-exon structure of the 5' is completely different. We agree with the reviewer's assessment, however, when it comes to identification of SNHG orthologs in more evolutionary distant species, closer to the root of the vertebrate clade.

Regarding splice sites, they are often conserved among lncRNA in general (gene structure conservation occurs more frequently than sequence conservation, see Ulitsky, Nat Rev Genet, 2016). In the case of SNHG7 the structure of the gene appears conserved in the mouse (though this annotation has likely not been fully confirmed experimentally), and in the monkey based on genome alignments. However, our RACE experiments show that the 5' end of SNHG7 in the monkey has a radically different splicing pattern when compared to human, so it is difficult to assess splicing conservation in the absence of full isoform characterization.

2) I am not perfectly convinced by the enrichment of miRNA target genes among the genes that are downregulated upon SNHG knockdown. The methods do not clearly explain how this enrichment is calculated. What is the background used for this enrichment analysis? In Figure 5C, we see that the genes predicted to be targeted by the top 10 microRNAs tend to have negative fold changes in the differential expression analysis (downregulation following knockdown). However, from Figure 5A it seems that the great majority of significantly differentially expressed (DE) genes have negative fold changes. How do the miRNA target genes differ from all other DE genes? What proportions of all predicted miRNA target genes (expressed in keratinocytes) are DE following knockdown, and how does this compare with the target genes of other miRNAs?

We have added a description of the statistical test used by MIENTURNET for the enrichment analysis to the methods section. More details can be found in the original publication. The significance of the enrichment is calculated by performing a hypergeometric test using as background the total number of miRNA-target interactions in the database and the total interactions the individual miRNA being tested engages in.

Figure 5C only includes the genes we used in our enrichment analysis (i.e. the significantly downregulated genes, not all DE genes) and it's meant to show the extent of downregulation exhibited by the targets of the most significantly enriched miRNAs within this group of genes.

The reviewer is correct in pointing out that the imbalance between downregulated and upregulated genes (which we now further highlight in the plots in Fig. 5A-B) will tend to skew any group of genes towards having a relatively large number of downregulated genes. However, we found this bias to be particularly strong in the case of our candidate miRNAs. We now show this in volcano plots for validated targets of the candidate miRNAs and a control miRNA (Fig. S8E). In a similar way, when looking at the cumulative distributions of the fold changes of all predicted targets for a certain miRNA and comparing it to the fold change cumulative distribution of all other genes, our candidate miRNAs displayed a more pronounced shift towards downregulation than the control miR-21-5p (which we now added in Fig. S8F).

Minor comments:

1) The AD (atopic dermatitis) abbreviation should be explained the first time it is mentioned in the text.

We thank the reviewer for pointing out the missing abbreviation, we have now added it.

2) More details are needed regarding the MIENTURNET analyses in the methods and in the main text

We have added more details about the statistics involved to the methods (see above).

3) Figure 5C, it is not clear how to interpret the color code for the boxplot. Does this represent the median or mean FDR of the target genes? Are only genes with FDR<5% included in this analysis?

The genes included in the enrichment analysis are all downregulated genes with an adjusted p-value (or FDR adjustment after the Wald test) < 0.05; in the manuscript we refer to this value as “ p_{adj} ”. The color scale in Fig 5C reports the significance of the enrichment for targets of the single miRNAs within the significantly downregulated genes list (FDR adjustment after the hypergeometric test), not the significance of the downregulation itself. FDR values are also used for all other enrichment analyses (GO terms, REACTOME Pathways). We apologise to the reviewer for the confusion, we have now modified the text and figures to make this clearer.

4) Figure 6D, I am not sure how to the D panel. Do the gray rectangles represent the exonic length of the SNHG? Do the dots correspond to the positions of the miRNA target sites? Here, a more quantitative comparison with the extent of sequence conservation of miRNA binding sites in SNHG, other lncRNAs and in protein-coding genes would be perhaps better suited.

The grey rectangles in Fig. 6D represent the total exonic length of the SNHG (basically all exons are “stitched together” head to tail irrespective of the actual isoforms) and the dots represent the positions of the miRNA binding sites within this “maximum exonic coverage”. Since not all individual isoforms are analysed it is possible that some additional miRNA sites can be created at alternatively spliced junctions, however we would estimate the number of such sites to be small. We have added this caveat to the methods section.

The degree of conservation that is estimated by TargetScan to underlie a functional MRE in coding genes is taken into account in this analysis, as sites within SNHG that pass this threshold are highlighted with yellow borders in the figure. We have now added a plot of the distribution of Branch length scores for MREs in SNHG and the distribution of Branch length scores for MREs in a random sample of 250 Coding genes UTRs (Fig. S9E). A similar comparison for lncRNA is more challenging as the data is not readily available and is likely to be confounded by the nuclear localisation of a majority of lncRNA species.

4. Description of analyses that authors prefer not to carry out

Please include a point-by-point response explaining why some of the requested data or additional analyses might not be necessary or cannot be provided within the scope of a revision. This can be due to time or resource limitations or in case of disagreement about the necessity of such additional data given the scope of the study. Please leave empty if not applicable.

Dear Fiona, dear Matteo,

Thank you for transferring your manuscript entitled "Neutral evolution of snoRNA Host Gene long non-coding RNA affects cell fate control" [EMBOJ-2024-117071-T; RC-2023-02280] to The EMBO Journal with Review Commons referee reports and responses to the concerns raised.

Given the referees' positive recommendations, and the overall interest of your findings, I would like to invite you to submit a revised version of the manuscript, addressing the comments of the reviewers along the lines sketched in your response letter.

As you know it is EMBO Journal policy to allow only a single round of revision, accordingly acceptance of your manuscript will therefore depend on the completeness of your responses in this revised version.

When preparing your letter of response to the referees' comments, please bear in mind that this will form part of the Review Process File, and will therefore be available online to the community. For more details on our Transparent Editorial Process, please visit our website.

Thank you for the opportunity to consider your work for publication. I look forward to your revision.

Best regards,

Daniel

Daniel Klimmeck, PhD
Senior Editor
The EMBO Journal

Instruction for the preparation of your revised manuscript:

- 1) a .docx formatted version of the manuscript text (including legends for main figures, EV figures and tables). Please make sure that the changes are highlighted to be clearly visible.
- 2) individual production quality figure files as .eps, .tif, .jpg (one file per figure).
- 3) a .docx formatted letter INCLUDING the reviewers' reports and your detailed point-by-point response to their comments. As part of the EMBO Press transparent editorial process, the point-by-point response is part of the Review Process File (RPF), which will be published alongside your paper.
- 4) a complete author checklist, which you can download from our author guidelines ([https://wol-prod-cdn.literatumonline.com/pb-assets/embo-site/Author Checklist%20-%20EMBO%20J-1561436015657.xlsx](https://wol-prod-cdn.literatumonline.com/pb-assets/embo-site/Author%20Checklist%20-%20EMBO%20J-1561436015657.xlsx)). Please insert information in the checklist that is also reflected in the manuscript. The completed author checklist will also be part of the RPF.
- 5) Please note that all corresponding authors are required to supply an ORCID ID for their name upon submission of a revised manuscript.
- 6) It is mandatory to include a 'Data Availability' section after the Materials and Methods. Before submitting your revision, primary datasets produced in this study need to be deposited in an appropriate public database, and the accession numbers and database listed under 'Data Availability'. Please remember to provide a reviewer password if the datasets are not yet public (see <https://www.embopress.org/page/journal/14602075/authorguide#datadeposition>). In case you have no data that requires deposition in a public database, please state so in this section. Note that the Data Availability Section is restricted to new primary data that are part of this study.
*** Note - All links should resolve to a page where the data can be accessed. ***
- 7) Our journal encourages inclusion of *data citations in the reference list* to directly cite datasets that were re-used and

obtained from public databases. Data citations in the article text are distinct from normal bibliographical citations and should directly link to the database records from which the data can be accessed. In the main text, data citations are formatted as follows: "Data ref: Smith et al, 2001" or "Data ref: NCBI Sequence Read Archive PRJNA342805, 2017". In the Reference list, data citations must be labelled with "[DATASET]". A data reference must provide the database name, accession number/identifiers and a resolvable link to the landing page from which the data can be accessed at the end of the reference. Further instructions are available at .

8) At EMBO Press we ask authors to provide source data for the main and EV figures. Our source data coordinator will contact you to discuss which figure panels we would need source data for and will also provide you with helpful tips on how to upload and organize the files.

Numerical data can be provided as individual .xls or .csv files (including a tab describing the data). For 'blots' or microscopy, uncropped images should be submitted (using a zip archive or a single pdf per main figure if multiple images need to be supplied for one panel). Additional information on source data and instruction on how to label the files are available at .

9) We replaced Supplementary Information with Expanded View (EV) Figures and Tables that are collapsible/expandable online (see examples in <https://www.embopress.org/doi/10.15252/embj.201695874>). A maximum of 5 EV Figures can be typeset. EV Figures should be cited as 'Figure EV1, Figure EV2' etc. in the text and their respective legends should be included in the main text after the legends of regular figures.

11) For data quantification: please specify the name of the statistical test used to generate error bars and P values, the number (n) of independent experiments (specify technical or biological replicates) underlying each data point and the test used to calculate p-values in each figure legend. The figure legends should contain a basic description of n, P and the test applied. Graphs must include a description of the bars and the error bars (s.d., s.e.m.).

We realize that it is difficult to revise to a specific deadline. In the interest of protecting the conceptual advance provided by the work, we recommend a revision within 3 months (6th Jun 2024). Please discuss the revision progress ahead of this time with the editor if you require more time to complete the revisions.

Use the link below to submit your revision:

Link Not Available

Rev_Com_number: RC-2023-02280

New_manu_number: EMBOJ-2024-117071-T

Corr_author: Watt

Title: Neutral evolution of snoRNA Host Gene long non-coding RNA affects cell fate control

We would like to sincerely thank both reviewers for taking the time to examine our work, for the cogent points they have raised, and their constructive attitude aimed at improving our manuscript.

Reviewer #1

1. Single-cell RNAseq (Fig. 1B) and in situ (Fig. 3D) results both indicate that SNHG7 is broadly expressed in multiple epidermal layers but more enriched in the spinous layer. Although most assays, such as colony formation and Ki67 staining, did not specifically examine the role of SNHG7 in the spinous layer, the raft culture experiment seemed to indicate specific reduction of the spinous layer (Fig. 3H), which was more prominent than basal defects. The authors should examine the defects more carefully in the raft culture system by using basal, spinous and granular markers. It is possible that SNHG7 functions to maintain limited cell proliferation while restrict premature differentiation. In addition, they should perform serial passage experiments to distinguish whether overexpression of SNHG7 can indeed confer self-renewal in long-term experiments.

We have included staining and quantification of a range of epidermal markers in the raft cultures (ITGB1, K14, K10 and IVL) in the revised manuscript (Fig. S6B). We do not observe major changes in the distribution of any of the differentiation markers. In terms of serial passage, we have cultured SNHG7-overexpressing or control cells for multiple passages until their growth capacity was exhausted. The SNHG7-overexpressing cells grew for approximately seven more passages than the control cells. We have added this information to the revised manuscript (Fig. S6L).

2. The main proposed mechanism is the sequestration of miR34 by SNHG7. While miR34 is well known for its function in inhibiting cell proliferation, the ability of coding or noncoding RNA to sequester miRNAs is highly dependent on the stability and copy number of these RNAs. Since they have single-cell data with UMI information, they should estimate the copy number of SNHG7 in epithelial cell populations, and this could provide a range for the "buffering" capacity of SNHG7. They should also examine, ideally by in situ hybridization, the expression patterns of miR34 in human vs mouse skin. While miR34 expression can be induced by p53 activation, it is possible that its expression varies in different species. It'll be interesting to determine whether the lack of miR34 expression in mouse keratinocyte or mouse skin could explain the insensitivity of mouse keratinocytes to SNHG7. Finally, to further demonstrate the competition between SNHG7 and miR34 targets, they can use a heterologous luciferase reporter system with a canonical miR34 targeting site in the 3'UTR and quantify luciferase activities with or without SNHG7 (or SNHG7 mut34 variant). This assay could quantify the impact of SNHG7 on individual miR34 targets.

We have now included an analysis of the estimated copy number of SNHG RNAs per cell based on the scRNAseq data (Fig.S3B). We also added the estimated copy number for known keratinocyte signalling and transcription factor mRNAs for reference.

We have added in situ hybridization staining for miR-34a-5p in human and mouse epidermis as well as mouse keratinocytes (Fig. S9D-F). We found miR-34a-5p to be expressed throughout the epidermal thickness in both species and to be highly expressed in cultured mouse cells. We performed heterologous luciferase reporter experiments in human keratinocytes where SNHG7 was knocked down (Fig. 6F) or in cells overexpressing wt SNHG7, or a miR-34-5p binding-deficient mutant or GFP only (Fig. 6G). We found that downregulation of SNHG7 significantly reduced luciferase reporter expression, while overexpression of wt SNHG7, but not the miR-34-5p MRE mutant, significantly increased luciferase activity. We have expanded our discussion in consideration of this new data.

Reviewer #2

1) The premise of the study relies on the observation that SNHG7s have low levels of sequence conservation. Indeed the authors aim to prove that biological functions can be identified even in the absence of evolutionary conservation. However, the extent of evolutionary conservation strongly depends on the phylogenetic scale at which it is analyzed. Here, the authors evaluate sequence conservation using PhastCons and PhyloP scores determined using alignments of human and 99 other vertebrate species. These scores reflect the extent of long-term sequence conservation. At this scale, only a small percentage of the human genome can be considered to be "conserved". It is thus not surprising that SNHG7 and other lncRNAs are not conserved at this scale, even if they carry some biological functions in the human genome. Here, it would be useful to redo the sequence conservation analyses using PhastCons and PhyloP scores computed on less distant species. Pre-computed scores exist for alignments containing human and other mammalian species, mainly primates (UCSC genome browser). It would also be good to provide comparisons of sequence conservation levels on the snoRNA genes and on the non-snoRNA parts of SNHG7s. In addition to protein-coding genes, pseudogenes and lncRNAs, it would be good to add a perfectly neutral control for the sequence conservation in Figure 1C - for example, flanking intergenic regions for SNHG7s. It might also be a good idea to analyze the GC content of SNHG7s compared to other lncRNAs, since GC content can be correlated with sequence conservation levels, in particular in noncoding regions. Importantly, the SNHG7 selected for detailed investigation (SNHG7) appears to be more conserved than the bulk of human lncRNAs, given that it is found in another primate and in mouse. It would be interesting to analyze in details what sequence features of this lncRNAs are conserved among species - for example, are the SNHG7 splice sites and promoter regions conserved? Are the snoRNA genes always located in the introns?

The reviewer raises very good points. We have added the conservation data of snoRNA genes (all or SNHG7-resident, both significantly more conserved than SNHG7s or lncRNA), and a "true neutral" control (we used the introns of 10,000 randomly sampled genes) to our PhastCons analysis (Fig.1C). We also added a conservation analysis of the promoters (defined as the 500 bp upstream of the transcription start site) of coding genes, pseudogenes, lncRNA and SNHG7s (Fig, S1B). We have now performed all our conservation analyses using both PhastCons scores generated from the 100-vertebrate alignment and PhastCons scores generated from the 30-mammals (28 primate) alignment (Fig. S1E-G). We do not detect any marked difference between the two alignment sets. We have also added a comparison of the GC content in lncRNA, SNHG7s, coding genes and introns (Fig.S1C).

Due to the nature of the PhyloP scores, the 30-mammal PhyloP track (phyloP30) would not be suitable to detect additional conservation in the primate lineage using the thresholding analysis we employed in Fig. 1E. The PhyloP track gives base-wise p-values for conservation (positive

values) or accelerated evolution (negative values). Using alignments of genomes that are overall more similar to each other (as in the 30-mammal alignment set) makes it more difficult to distinguish between conserved and neutrally evolving regions, because even segments that are not under constraint will look relatively similar due to the evolutionary proximity of the species in the set. For the same reason this alignment set is quite sensitive to accelerated evolution, as it contains many relatively similar genomes in the alignment.

As a result, the PhyloP30 scores are very asymmetrical around zero: the conservation (positive) scores never reach 2 (p -value of 0.01) in the whole track (not even coding regions of very well-conserved genes), while acceleration scores can reach very significant values, down to -20. Conversely, the PhyloP100 track (used in Fig.1E) is quite symmetrical around 0 and is thus better suited for the purposes of the analysis in Fig.1E, which we used to detect both conserved and accelerated portions of SNHG7. We did, however, inspect the PhyloP30 track manually and did not observe any clear evidence of the presence of additional conserved elements in SNHG7. We have added all conservation tracks for SNHG7 to Fig. 3A.

While lncRNA orthologs can be identified using a combination of sequence conservation, conserved synteny with surrounding genes and in some cases conserved gene structure, SNHG7 orthologs can additionally be identified by the embedded (conserved) intronic snoRNA sequence, which makes them easier to find even when the transcript sequence bears no similarity across species. The mouse SNHG7 sequence, for example, does not match with human SNHG7 even using the least stringent BLAST parameters. The monkey sequence is similar enough to match with the human in the 3' of the gene, but the intron-exon structure of the 5' is completely different. However, as the reviewer implies, even with the help of the snoRNA sequence, identification of SNHG7 orthologs in more evolutionarily distant species – closer to the root of the vertebrate clade – may be more challenging and we now consider this in the interpretation of some of our results in the main text.

Regarding splice sites, they are often conserved among lncRNA in general (gene structure conservation occurs more frequently than sequence conservation; see Ulitsky, Nat Rev Genet, 2016). In the case of SNHG7 the structure of the gene appears conserved in the mouse (though this annotation has likely not been fully confirmed experimentally), and in the monkey based on genome alignments. However, our RACE experiments show that the 5' end of SNHG7 in the monkey has a radically different splicing pattern when compared to human, so it is difficult to assess splicing conservation in the absence of full isoform characterization.

2) I am not perfectly convinced by the enrichment of miRNA target genes among the genes that are downregulated upon SNHG7 knockdown. The methods do not clearly explain how this enrichment is calculated. What is the background used for this enrichment analysis? In Figure 5C, we see that the genes predicted to be targeted by the top 10 microRNAs tend to have negative fold changes in the differential expression analysis (downregulation following knockdown). However, from Figure 5A it seems that the great majority of significantly differentially expressed (DE) genes have negative fold changes. How do the miRNA target genes differ from all other DE genes? What proportions of all predicted miRNA target genes (expressed in keratinocytes) are DE following knockdown, and how does this compare with the target genes of other miRNAs?

We have added a description of the statistical test used by MIENTURNET for the enrichment analysis to the methods section. More details can be found in the original publication. The significance of the enrichment is calculated by performing a hypergeometric test using as background the total number of miRNA-target interactions in the database and the total interactions the individual miRNA being tested engages in.

Figure 5C only includes the genes we used in our enrichment analysis (i.e. the significantly downregulated genes, not all DE genes) and is meant to show the extent of downregulation exhibited by the targets of the most significantly enriched miRNAs within this group of genes. The reviewer is correct in pointing out that the imbalance between downregulated and upregulated genes (which we now further highlight in the plots in Fig. 5A-B) will tend to skew any group of genes towards having a relatively large number of downregulated genes. However, we found this bias to be particularly strong in the case of our candidate miRNAs. We now show this in volcano plots for validated targets of the candidate miRNAs and a control miRNA (Fig. S8E). In a similar way, when looking at the cumulative distributions of the fold changes of all predicted targets for a certain miRNA and comparing it to the fold change cumulative distribution of all other genes, our candidate miRNAs displayed a more pronounced shift towards downregulation than the control miR-21-5p (which we have now added in Fig. S8F).

3) If keratinocyte RNA-seq data is available for other species (for example mouse), it would be interesting to test whether the high expression levels of SNHG7 and the other analyzed SNHGs are also conserved in the other species.

We have now included a comparison of the gene expression levels of *Snhgs* relative to other transcript classes in mouse (Fig. S1C). We find a similar expression pattern to that observed in humans, indicating that the generally high level of expression of snoRNA host genes is a conserved feature.

Minor comments:

1) The AD (atopic dermatitis) abbreviation should be explained the first time it is mentioned in the text.

We thank the reviewer for pointing out the missing abbreviation; we have now added it.

2) More details are needed regarding the MIENTURNET analyses in the methods and in the main text

We have added more details as requested (see above).

3) Figure 5C, it is not clear how to interpret the color code for the boxplot. Does this represent the median or mean FDR of the target genes? Are only genes with FDR<5% included in this analysis?

The genes included in the enrichment analysis are all downregulated genes with an adjusted p-value (or FDR adjustment after the Wald test) < 0.05 ; in the manuscript we refer to this value as " p_{adj} ". The color scale in Fig 5C reports the significance of the enrichment for targets of the single miRNAs within the significantly downregulated genes list (FDR adjustment after the hypergeometric test), not the significance of the downregulation itself. FDR values are also used for all other enrichment analyses (GO terms, REACTOME Pathways). We apologise to the reviewer for the confusion and have now modified the text and figures to make this clearer.

4) Figure 6D, I am not sure how to the D panel. Do the gray rectangles represent the exonic length of the SNHGs? Do the dots correspond to the positions of the miRNA target sites? Here,

a more quantitative comparison with the extent of sequence conservation of miRNA binding sites in SNHG_s, other lncRNAs and in protein-coding genes would be perhaps better suited.

The grey rectangles in Fig. 6D represent the total exonic length of the SNHG_s (i.e. all exons are “stitched together” head to tail irrespective of the actual isoforms) and the dots represent the positions of the miRNA binding sites within this “maximum exonic coverage”. Since not all individual isoforms are analysed, it is possible that some additional miRNA sites can be created at alternatively spliced junctions, although we estimate the number of such sites to be small. We have added this caveat to the methods section.

The degree of conservation that is estimated by TargetScan to underlie a functional MRE in coding genes is taken into account in this analysis, as sites within SNHG_s that pass this threshold are highlighted with yellow borders in the figure. We have now added a plot of the distribution of Branch length scores for MREs in SNHG_s and the distribution of Branch length scores for MREs in a random sample of 250 Coding genes UTRs (Fig. S9E). A similar comparison for lncRNA is more challenging as the data is not readily available and is likely to be confounded by the nuclear localisation of a majority of lncRNA species.

Dear Dr Watt, dear Dr Vietri Rudan,

Thank you for submitting your revised manuscript (EMBOJ-2024-117071R) to The EMBO Journal. Your amended study was sent back to the two referees for their scientific evaluation, and we have received detailed comments from both of them, which I enclose below. As you will see, the experts state that the work has been substantially improved by the revisions and they are now in favour of publication, pending minor revision.

Thus, we are pleased to inform you that your manuscript has been accepted in principle for publication in The EMBO Journal.

Please consider the remaining minor concerns of referee #1 regarding composition of the reporter and statistical analysis carefully and amend the manuscript with complementary experiments and text accordingly where appropriate.

Also, we now need you to take care of a number of issues related to formatting and data presentation as detailed below, which should be addressed at re-submission.

Please contact me at any time if you have additional questions related to below points.

As you might have seen on our web page, every paper at the EMBO Journal now includes a 'Synopsis', displayed on the html and freely accessible to all readers. The synopsis includes a 'model' figure as well as 2-5 one-short-sentence bullet points that summarize the article. I would appreciate if you could provide this figure and the bullet points.

Thank you for giving us the chance to consider your manuscript for The EMBO Journal. I look forward to your final revision.

Again, please contact me at any time if you need any help or have further questions.

Best regards,

Daniel Klimmeck

>> Author Contributions: Please remove the author contributions information from the manuscript text. Note that CRediT has replaced the traditional author contributions section as of now because it offers a systematic machine-readable author contributions format that allows for more effective research assessment. and use the free text boxes beneath each contributing author's name to add specific details on the author's contribution.

More information is available in our guide to authors.

>> Introduce a 'Disclosure and Competing Interests Statement', adding "The authors declare that they have no conflict of interest. Fiona Watt is the director of EMBO. The EMBO Journal is editorially independent of EMBO."

>> Callouts: Fig. 4C is called out after Fig. 5 and Fig. 6; please correct.

>> Dataset EV legends: Supplementary Tables 1 - 5 can be removed as the information is already provided as part of the source data. References to the files in the text will need to be updated accordingly.

>> Appendix: The appendix file will need a table of content (please add page numbers) and the names of the figures will need to be corrected to "Appendix Figure S1, S2..etc". A more detailed description is needed in the legends of Appendix Figure S2 and S5.

>> Please add a completed Author Checklist to your manuscript.

- >> Add a separate 'Statistical Analysis' section to the manuscript, detailing the algorithms and tests applied.
- >> Detail animal housing and welfare considerations for your study in the Methods section (primary mouse and monkey cells).
- >> Consider additional changes and comments from our production team as indicated below:

- Figure legends:

Please note that a separate 'Data Information' section is required in the legends of figures 3b-i; 5a-b, d-e, g; 6b, d-g.

Revision to The EMBO Journal should be submitted online within 90 days, unless an extension has been requested and approved by the editor; please click on the link below to submit the revision online before 2nd Sep 2024:

Link Not Available

Referee #1:

In the revised manuscript, while the authors performed additional analysis and experiments, my concerns are not fully addressed. My major concern is their heterologous reporter assay showed very mild repression (<20%) of the miR34 reporter upon SNHG7 silencing. In contrast, miR34 mimic showed a very strong effect (>80% repression) as expected. There are two potential issues with the experiment. First, the choice of the miR34 reporter is a poor decision. This reporter contains an artificial, fully complementary miR34 target site, which should be cleaved by miR34 and repressed much more strongly by miR34 than typical miR34 targets. If such a strong target (usually not found in any endogenous miRNA target) is only repressed mildly (less than 20%) upon SNHG7 silencing and derepressed mildly (less than 20%) upon SNHG7 over expression (Fig. 6G), I'm not convinced endogenous targets will show any noticeable response. They should use a reporter with an endogenous miR34 target site, such as CCNE2, CDK4 (He L, Nature 2007) or any relevant targets in keratinocytes, to evaluate the functional interference of SNHG7 on miR34. The second issue is that the p value for the difference between siScramble and siSNHG7 ($p < 0.0001$ in Fig. 6F) seems implausible given the minor difference and relatively large variations, and the authors should double check.

For their estimation of RNA count, they should provide a description for the method of quantification.

Referee #2:

I am satisfied with the authors' reply to my initial comments. I have no further remarks and recommend publication.

Rev_Com_number: RC-2023-02280

New_manu_number: EMBOJ-2024-117071R

Corr_author: Watt

Title: Neutral evolution of snoRNA Host Gene long non-coding RNA affects cell fate control

Reviewer #1

In the revised manuscript, while the authors performed additional analysis and experiments, my concerns are not fully addressed. My major concern is their heterologous reporter assay showed very mild repression (<20%) of the miR34 reporter upon SNHG7 silencing. In contrast, miR34 mimic showed a very strong effect (>80% repression) as expected. There are two potential issues with the experiment. First, the choice of the miR34 reporter is a poor decision. This reporter contains an artificial, fully complementary miR34 target site, which should be cleaved by miR34 and repressed much more strongly by miR34 than typical miR34 targets. If such a strong target (usually not found in any endogenous miRNA target) is only repressed mildly (less than 20%) upon SNHG7 silencing and derepressed mildly (less than 20%) upon SNHG7 over expression (Fig. 6G), I'm not convinced endogenous targets will show any noticeable response. They should use a reporter with an endogenous miR34 target site, such as CCNE2, CDK4 (He L, Nature 2007) or any relevant targets in keratinocytes, to evaluate the functional interference of SNHG7 on miR34. The second issue is that the p value for the difference between siScramble and siSNHG7 ($p < 0.0001$ in Fig. 6F) seems implausible given the minor difference and relatively large variations, and the authors should double check.

For their estimation of RNA count, they should provide a description for the method of quantification.

We thank the reviewer for their careful analysis. Our aim with the luciferase assay was to show that transcripts that are sensitive to miR-34 regulation can be affected by SNHG7 levels. Our approach favoured simplicity and clarity over attempting to emulate endogenous conditions, because we wanted to avoid the presence of any confounding factors. The construct we selected was used in recent literature, it had a simple structure with a short 3'UTR that minimised the potential influence of other sequences, and it showed good sensitivity. A recent comparison of individual miR-34 MREs shows that while full complementarity indeed confers the strongest responsiveness to miR-34 maximal repression and an increased affinity to Ago2/miR-34 complexes, these measurements remained within the same order of magnitude as "strong" endogenous sites (Sweetapple et al., 2024). It should also be considered that endogenous transcripts can contain multiple MREs that can act additively or even synergistically (Grimson et al., 2007), while the construct we used only contains one. For these reasons, we do expect (and indeed observe) the luciferase transcript to be strongly targeted by miR-34, but not aberrantly so. In addition, our RNAseq data shows that the level of downregulation of miR-34 targets in SNHG7 knockdown keratinocytes does not seem to correlate with the predicted strength of miRNA targeting on each transcript (we have added this analysis to Appendix Fig. S8G). We agree with the reviewer that these experimental caveats should be mentioned and we have added these considerations to the manuscript.

We have re-checked that the p-value for the difference between siScramble and siSNHG7 ($p < 0.0001$ in Fig. 6F) is correct.

We have added more detail on the extraction of estimated RNA counts from the scRNAseq data to the Methods section.

Dear Dr Watt,

Thank you for submitting the revised version of your manuscript. I have now evaluated your amended manuscript and concluded that the remaining minor concerns have been sufficiently addressed.

I am thus pleased to inform you that your manuscript has been accepted for publication in the EMBO Journal.

On a different note, I would like to alert you that EMBO Press offers a format for a video-synopsis of work published with us, which essentially is a short, author-generated film explaining the core findings in hand drawings, and, as we believe, can be very useful to increase visibility of the work. Please see the following link for representative examples and their integration into the article web page:

<https://www.embopress.org/doi/full/10.15252/emj.2019103932>

Finally, we have noted that the submitted version of your article is also posted on the preprint platform bioRxiv. We would appreciate if you could alert bioRxiv on the acceptance of this manuscript at The EMBO Journal in order to allow for an update of the entry status. Thank you in advance!

Best regards,

Daniel Klimmeck

Daniel Klimmeck, PhD
Senior Editor
The EMBO Journal
EMBO
Postfach 1022-40
Meyerohofstrasse 1
D-69117 Heidelberg
contact@embojournal.org

Submit at: <http://emboj.msubmit.net>

Rev_Com_number: RC-2023-02280

New_manu_number: EMBOJ-2024-117071R1

Corr_author: Watt

Title: Neutral evolution of snoRNA Host Gene long non-coding RNA affects cell fate control